# Systemic *Helicobacter* infection and associated mortalities in endangered Grand Cayman blue iguanas (*Cyclura lewisi*) and introduced green iguanas (*Iguana iguana*)

Kenneth J. Conley[1][☯]*, Tracie A. Seimon[1][☯], Ioana S. Popescu[2][¤a], James F. X. Wellehan, Jr.[3], James G. Fox[4,5], Zeli Shen[4], Jane Haakonsson[6], Anton Seimon[7], Ania Tomaszewicz Brown[1][¤b], Veronica King[8][¤c], Fred Burton[6], Paul P. Calle[1]

1 Wildlife Conservation Society, Zoological Health Program, Bronx, New York, United States of America, 2 Island Veterinary Services, George Town, Cayman Islands, 3 Department of Comparative, Diagnostic & Population Medicine, University of Florida, Gainesville, Florida, United States of America, 4 Division of Comparative Medicine, Massachusetts Institute of Technology, Cambridge, Massachusetts, United States of America, 5 Department of Biological Engineering, Massachusetts Institute of Technology, Cambridge, Massachusetts, United States of America, 6 Department of Environment, Cayman Islands Government, Grand Cayman, Cayman Islands, 7 Center for Environmental Policy, Bard College, Annandale-on-Hudson, New York, United States of America, 8 St. Matthew's University, School of Veterinary Medicine, Grand Cayman, Cayman Islands

☯ These authors contributed equally to this work.
¤a Current address: Newmarket Vets4Pets, Newmarket, United Kingdom
¤b Current address: The European Association of Zoos and Aquaria, Amsterdam, Netherlands
¤c Current address: Lincoln-Memorial University, College of Veterinary Medicine, Harrogate, Tennessee, United States of America
* kconley@wcs.org

## Abstract

The Blue Iguana Recovery Programme maintains a captive breeding and head-starting program for endangered Grand Cayman blue iguanas (*Cyclura lewisi*) on Grand Cayman, Cayman Islands. In May 2015, program staff encountered two lethargic wild Grand Cayman blue iguanas within the Queen Elizabeth II Botanic Park (QEIIBP). Spiral-shaped bacteria were identified on peripheral blood smears from both animals, which molecular diagnostics identified as a novel *Helicobacter* species (provisionary name *Helicobacter* sp. GCBI1). Between March 2015 and February 2017, 11 Grand Cayman blue iguanas were identified with the infection. Two of these were found dead and nine were treated; five of the nine treated animals survived the initial infection. Phylogenetic analysis of the 16S rRNA gene suggests *Helicobacter* sp. GCBI1 is most closely related to *Helicobacter* spp. in chelonians. We developed a Taqman qPCR assay specific for *Helicobacter* sp. GCBI1 to screen tissue and/or blood samples from clinical cases, fecal and cloacal samples from clinically healthy Grand Cayman blue iguanas, including previously infected and recovered iguanas, and iguanas housed adjacent to clinical cases. Fecal and/or cloacal swab samples were all negative, suggesting that Grand Cayman blue iguanas do not asymptomatically carry this organism nor shed this pathogen per cloaca post infection. Retrospective analysis of a 2014 mortality event affecting green iguanas (*Iguana iguana*) from a separate Grand Cayman location identified *Helicobacter* sp. GCBI1 in two of three cases. The source of infection and

**Data Availability Statement:** All relevant data are within the manuscript and its Supporting information files.

**Funding:** Funding for this project was provided by the Wildlife Conservation Society's Zoological Health Program, the Cayman Islands Government, the National Trust for the Cayman Islands, the Derald H Ruttenberg Memorial Fund (PPC), and the following grants (JGF): P30-ES0002109, R01-OD011141, and T32-OD010978. One author (ISP) was commercially employed through the duration of the study and manuscript preparation (Island Veterinary Services and Newmarket Vets4Pets); funding was limited to salary. None of the funders had any role in study design, data collection and analysis, decision to publish, or preparation of the manuscript. The specific roles of all authors are articulated in the "author contributions" section.

**Competing interests:** I have read the journal's policy and the authors of this manuscript have the following competing interests: One author (ISP) was commercially employed through the duration of the study and manuscript preparation (Island Veterinary Services and Newmarket Vets4Pets); funding from these businesses was limited to salary support. This does not alter our adherence to PLOS ONE policies on sharing data and materials.

mode of transmission are yet to be confirmed. Analysis of rainfall data reveal that all infections occurred during a multi-year dry period, and most occurred shortly after the first rains at the end of seasonal drought. Additionally, further screening has identified *Helicobacter* sp. GCBI1 from choanal swabs of clinically normal green iguanas in the QEIIBP, suggesting they could be asymptomatic carriers and a potential source of the pathogen.

## Introduction

Grand Cayman blue iguanas (*Cyclura lewisi*) are large, ground-dwelling iguanas endemic to Grand Cayman, Cayman Islands. Closely related to several other rock iguana species distributed throughout the Caribbean, they exhibit a unique blue coloration, most prominently in healthy adult males. Once thought to number into the tens of thousands, by the 1920s, they were considered scarce, and in 1990 the population was estimated to consist of 100–200 iguanas [1]. One decade later, the population was considered functionally extinct, with only 10–25 animals remaining in the wild. Intensive conservation efforts began in 2002 with the creation of the Blue Iguana Recovery Programme (BIRP) [1]. The BIRP is operated by the National Trust for the Cayman Islands in partnership with the Cayman Islands Department of Environment and a range of international organizations. It maintains a captive breeding, head-starting and release facility within the Queen Elizabeth II Botanic Park (QEIIBP) on Grand Cayman (Fig 1). Over the last 30 years, the BIRP has incrementally restored wild breeding populations of Grand Cayman blue iguanas in the QEIIBP, and in two National Trust for the Cayman Islands protected areas in eastern Grand Cayman, the Salina Reserve and the Colliers Wilderness Reserve. The Wildlife Conservation Society's (WCS) Zoological Health Program has provided veterinary support for the BIRP since 2001, in conjunction with the IUCN Iguana Specialist Group, and with assistance from the Saint Matthew's University School of Veterinary Medicine. Activities have included pre-release evaluations, health assessments, and annual examinations of Grand Cayman blue iguanas at the breeding facility, as well as free-ranging iguanas [2–4].

Green iguanas were introduced to western Grand Cayman and became evident in the wild in the late 1980s and early 1990s. This species proved to be highly invasive, and by 2017 had reached an estimated population size on Grand Cayman of 1.3 million individuals, distributed island-wide but with highest density in human-modified habitats throughout the island's western districts [5]. Due to this overabundance, culling efforts have been underway since October 2018 as a measure to keep the green iguana population under control [5].

*Helicobacter* species are Gram negative, microaerophilic, and fastidious bacteria in the phylum Proteobacteria, class Epsilonproteobacteria, order Campylobacterales, and can cause disease in a wide array of vertebrate hosts, including reptiles. They can be curved, helical or fusiform [6]. *Helicobacter* species roughly group phylogenetically into those of gastric origin and those of enterohepatic origin. Diseases associated with helicobacteriosis in humans and animals include chronic gastritis, gastroduodenal ulcers, gastric carcinoma, enterocolitis, cellulitis, and sepsis [6]. The prevalence of *Helicobacter* spp. in reptiles can range from 4.8–39.1% [7], and both gastric (in Squamata) and enterohepatic (in Testudines) lineages of *Helicobacter* species have been identified [8]. *Helicobacter* species are thought to be a natural component of the reptilian microbiome and rarely associated with disease [8]. However, fatal septicemia associated with spiral organisms has been reported in a pancake tortoise (*Malacochersus tornieri*) in which a *Helicobacter* sp. was confirmed by molecular techniques [9], and in a rhinoceros iguana (*Cyclura cornuta*) in which taxonomic designation of the organism was not determined [10, 11].

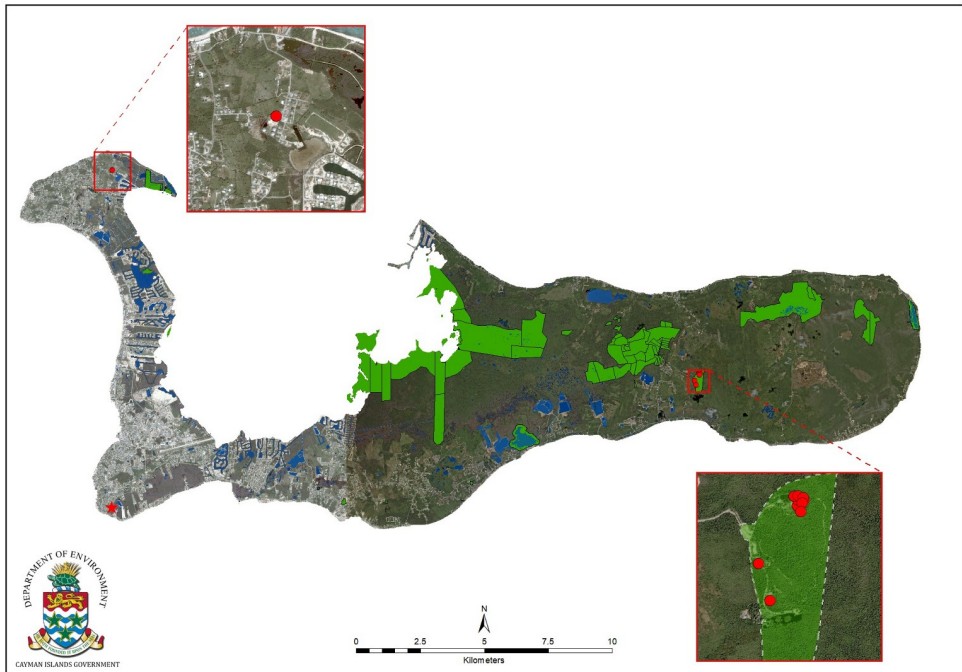

**Fig 1. Map of Grand Cayman, Cayman Islands.** Red foci represent confirmed *Helicobacter* positive cases. The Queen Elizabeth II Botanic Park (QEIIBP), where all positive Grand Cayman blue iguanas were found, is encircled by the dashed line in the right inset. The cluster of cases in the north of the QEIIBP is in and around the breeding facility. The West Bay region in the upper left inset is where confirmed *Helicobacter* positive cases in green iguanas were found and the red star in the southwest indicates the George Town location of additional green iguana mortalities not confirmed to be *Helicobacter* positive. Green regions in the main map represent protected areas and blue is inland water. Created by the Department of Environment, Cayman Islands Government. 2006 Quickbird satellite imagery courtesy of the Department of Environment, Cayman Islands Government.

In 2015, two free-ranging, clinically ill Grand Cayman blue iguanas were diagnosed with spiral-shaped bacteria circulating in the peripheral blood. Through retrospective and contemporaneous investigation into the resulting and ongoing mortality events in the endangered Grand Cayman blue iguanas, here we describe the clinical, pathological, and molecular presentation of a novel *Helicobacter* species and associated disease. This includes retrospective testing of, and identification of the disease in, introduced green iguanas in Grand Cayman, the development of a novel *Helicobacter*-specific PCR assay, identification of the organism in apparently healthy free-ranging green iguanas in Grand Cayman, and analysis of Grand Cayman climate data relative to infections.

## Materials and methods

### Ethics statement

IACUC approval for this project was not required as WCS institutional requirements for IACUC review do not include field projects that take place outside of our facilities. However, sampling, capture and release of wild reptiles, collecting dead animals found in the environment, or sampling after euthanasia are all standard techniques and procedures for clinical and pathology examinations or investigations. The welfare of animals included in this study was considered throughout their care, with analgesics used as deemed necessary. Animals that survived infection were released back into the environment where they were found, either in the captive setting or free-ranging within the QEIIBP. No animals were housed for continued

research purposes. Within the Cayman Islands regulatory framework, the project operates under the terms of a protected species permit issued to the BIRP by the National Conservation Council. Blood, tissue, and fecal samples were collected by or under the direction of licensed veterinarians. Cayman Islands CITES export permits 2014/KY/000674, 2014/KY/000687, 2014/KY/000686, 2014/KY/000690, 2014/KY/000689, 2015/KY/000777, 2015/KY/000808, 2015/KY/000809, 2016/KY/000817, 2017/KY/000874, and 2017/KY/000923; and United States CITES import permits 15US033594/9, 16US0333594/9, and 17US033594/9 were used for sample export and import.

## Case designation

Any case with at least one PCR positive result for the target *Helicobacter* sp. or with circulating spiral-shaped bacteria on peripheral blood smear was considered *Helicobacter* positive. The negative disease state was not strictly defined due to the inconsistency in sample availability between cases. Grand Cayman blue iguanas included in this review were either *Helicobacter* positive, or were found dead on Grand Cayman and a complete necropsy, including histology, was performed; cases included were presented from 2015 through 2017 and represented both free-ranging and captive individuals. All green iguanas from a mortality event in 2014 that were received for necropsy were also included. All individual blue iguanas are identified by their studbook number, and the green iguanas are identified by the necropsy number assigned to them by WCS.

## Climatology

Daily level precipitation data was obtained from two sites on Grand Cayman. Daily rainfall records from January 2003 through April 2018 were available from Queen Elizabeth II Botanic Park. Data outages occurred in several years (81.4% of all days have available data); however, for the 2014–16 study period, the records are complete. Observations from Owen Roberts International Airport, 20 km WSW of the QEIIBP, from January 1976 through October 2019, were provided by the Cayman Islands National Weather Service. Other than 15 dates missing in 2004 and six in 2005, the records are continuous (99.87%). To determine if *Helicobacter* infection of Grand Cayman blue iguanas is associated with precipitation, the date of onset of clinical signs or the date the animal was found dead was compared to records of daily precipitation.

## Clinical

To assess and define a clinical tableau for this disease, the medical records of all nine *Helicobacter* positive Grand Cayman blue iguanas that presented to a Cayman Islands veterinary hospital (Island Veterinary Services) between May 2015 and May 2017 were reviewed and analyzed. Recordings of history and physical examination notes, results of blood smears, biochemistry and complete blood counts, and imaging studies were analyzed.

Blood was collected via ventral tail vein venipuncture from all nine iguanas, and hematology smears were prepared using a wedge technique, stained using JorVet Dip Quick Stain (Jorgensen Labs, Loveland, CO, USA), and air dried. The volume of blood collected was not specified in the reviewed medical records; generally, the sample volume depends on the diagnostic testing expected to be performed, but does not exceed 2% of the blood volume. Blood smears were performed on blood collected on the day of presentation in all nine cases, and in three cases blood smears were repeated during follow up. The presence of circulating spiral-shaped bacteria was assessed by at least one of the authors (ISP and/or KJC) in all but one case

(Studbook No. 819). The blood smear of that individual was submitted to a commercial veterinary reference laboratory (IDEXX Operations, Memphis, TN, USA).

Blood was additionally preserved in lithium heparin and serum vials (both BD Vacutainer®, BD Life Science, Franklin Lakes, NJ, USA). Whole blood from three cases was submitted to a commercial veterinary reference laboratory for complete blood counts (IDEXX Operations). The serum vials were spun, and the serum was separated. Biochemistry testing was performed either with a portable analyzer (VETSCAN® VS2 Chemistry Analyzer, Abaxis Inc., Union City, CA, USA) using whole blood, or by a commercial veterinary reference laboratory (IDEXX Operations) utilizing serum. Biochemistry values were obtained from all nine animals from blood collected on admission, and three cases received follow up testing. Packed cell volume (PCV) was determined in-house using an electrical impedance automated hematology analyzer (IDEXX VetAutoread Hematology analyzer, IDEXX Laboratories Inc., Westbrook, ME, USA).

Radiographs of all patients were taken using a MinXray HF 100+ digital radiograph (MinXray, Northbrook, IL, USA). In a single case, coelomic ultrasound was performed using an Esaote MyLab Five VET (Esaote Group, Genoa, Italy).

Investigation into the 2014 green iguana mortality event was limited to field observation and postmortem examination, and antemortem diagnostics were not performed. In 2019, in conjunction with the Cayman Islands Department of Environment's green iguana culling program [5], choanal and cloacal swabs were collected from 15 culled, apparently healthy, free-ranging green iguanas at the QEIIBP. No additional diagnostics were performed in these animals.

## Pathology

Necropsies on dead iguanas were performed by a clinical veterinarian or a veterinary pathologist, or by students under the observation of a veterinary pathologist. When necropsies could not be performed in a timely manner, carcasses were frozen prior to examination. Tissues were collected in 10% neutral buffered formalin, or were frozen, and exported to the WCS Pathology Department once or twice annually. Collected tissue sets were inconsistent, but nearly all cases included at least liver, lung, kidney, stomach, intestine and skeletal muscle in formalin (one case, Studbook No. 3003 was markedly autolyzed and only skeletal muscle and skin were collected); central nervous system tissues were infrequently included. Frozen tissue collection varied case-to-case. Formalinized tissues were trimmed and processed routinely at the WCS (Shandon Excelsior™ ES, Thermo Scientific, Kalamazoo, MI, USA). Four to five micrometer thick sections were cut from paraffin blocks (RM2255 microtome, Leica, Bannockburn, IL, USA) and stained with hematoxylin and eosin (H&E) on an automatic stainer (Shandon Varistain® Gemini ES, Thermo Scientific, Kalamazoo, MI, USA). A modified Warthin-Starry stain was used for detection of *Helicobacter* spp. Briefly, deparaffinized slides were exposed to a spirochete sensitizer (zinc formalin) for one hour at 60˚ C, 1% silver nitrate solution for one hour at 60˚ C, gum mastic for five minutes at room temperature, and a developer/reducing solution (4.2 ml of 2% silver nitrate, 10.3 ml of 5% gelatin and 5.5 ml of 15% hydroquinone) in a 60˚ C water bath until yellow to golden brown color developed. Slides were washed with distilled water and/or alcohol between steps.

Images were obtained using an Olympus® Bx40 microscope, Olympus® DP71 camera, and Olympus® cellSens Standard software (v. 1.11).

## Molecular diagnostics

DNA was extracted from blood (both antemortem and postmortem) and frozen necropsy tissue using the High Pure PCR Template Preparation Kit (Roche Molecular Biochemicals,

Indianapolis, IN, USA) or QIAmp® DNA kit (Qiagen Inc., Valencia, CA, USA) according to the manufacturer's protocols. DNA was also extracted from 10 X 5.0 μm sections (scrolls) of formalin fixed-paraffin embedded necropsy tissue using the QIAmp® DNA FFPE Tissue Kit according to the manufacturer's protocol using deparaffinization solution (Qiagen Inc., Valencia, CA, USA). Fecal, choanal, and cloacal swab samples were collected into individual tubes containing 250 μl RNA*later*® (Life Technologies, Grand Island, NY, USA). All non-formalin-fixed samples were maintained frozen (-20˚C) from date of collection through exportation to WCS's Bronx Zoo, NY, USA, and then maintained at -80˚C until PCR analysis. Swab and fecal DNA was extracted using the QIAmp® DNA minikit or QIAmp® DNA stool kit, respectively (Qiagen Inc., Valencia, CA, USA). Non-frozen necropsy samples were collected and maintained in 10% neutral buffered formalin prior to histologic processing, which typically occurred several months after collection.

PCR was performed using previously described methods with either *Helicobacter*-specific [12] or pan-bacterial [13] protocols targeting the 16S rRNA gene. Amplified products were either subcloned and sequenced, or directly sequenced in the forward and reverse direction. All sequences were analyzed, trimmed of their primer sequences, and aligned to generate a consensus sequence, which was queried against available sequences in GenBank (National Center for Biotechnology Information, Bethesda, MD, USA).

To obtain longer sequence from a subset of positive samples for phylogenetic analysis, DNA was amplified using the following primers targeting the 16S rRNA gene: 16S-Forward, 5'-ATGGAGAGTTTGATCCTGGCT-3' and 16S-Reverse, 5'-ATCGGYTACCTTGTTACGAC TTC-3'. Positive PCR products were enzymatically treated with ExoSAP-IT® (Affymetrix, Santa Clara, CA, USA), and sequenced in both the forward and reverse directions (Eton Bioscience, Union, NJ, USA) using the following sequencing primers: 5'-CCAGCAGCCGCGGTA ATACG-3'; 5'-ATGGAGAGTTTGATCCTGGCT-3'; 5'-GAGTACGGTCGCAAGATTAAA ACTC-3'; 5'-ATCGGYTACCTTGTTACGACTTC-3'; 5'-GAGTTTTAATCTTGCGACCGTA CTC-3'.

Sequences were analyzed, trimmed and aligned using Geneious software (Geneious Pro R10, Biomatters LTD., Auckland, NZ; Geneious alignment tool with default settings) and sequence comparisons were performed using the GenBank database and BLASTn.

Phylogenetic analysis using FastTree approximate maximum-likelihood with 1000 bootstrap resamples (FastTree 2.1.5 plugin in Geneious Pro, using a GTR substitution model optimized for gamma20 likelihood) was performed to determine the relationship of the *Helicobacter* sp. from the Grand Cayman blue iguana to other *Helicobacter* spp. sequences in GenBank [14]. The tree was finalized and labeled using FigTreeV1.3.1 software (2006–2009, Andrew Rambaut; Institute of Evolutionary Biology, University of Edinburgh, Edinburgh, UK).

For qPCR testing, we developed a Taqman qPCR assay specific for *Helicobacter* sp. Grand Cayman Blue Iguana 1 (GCBI1; provisional name). The forward and reverse primers (GCBI-Helico-F, 5'-TAGATAACATGCCCTTTAGTCTGGGATAGCCA-3'; and GCBI-Helico-R, 5'-GTGTGTCCGTTCACCCTCTCAGG-3') amplified a 194bp region, and the Taqman probe targeted a 14bp insertion unique to *Helicobacter* sp. GCBI1 (Taqman probe, 5'-6FAM-CTGGATACTCCCCAAGGGGATACC-MGBNFQ-3'). Twenty μl reactions containing 10 μl of 2X Taqman Environmental Master Mix, 900 nM of each primer, 250 nM of probe, 2.0 μl of 10X exogenous internal positive control primers and probe, 0.4 μl of 50X exogenous internal positive control DNA, DNase/RNase-free water (all from Thermo Fisher Scientific, Waltham, MA, USA) and 2 μl of template DNA were added to each well. The exogenous internal positive control reagents served as inhibition controls in the PCR reactions, and no inhibition was found in any of the samples tested except for Studbook No. 819 (formalin-fixed, paraffin-embedded spleen). Because of the PCR inhibition observed with this sample, we

diluted the sample 1:10 and retested. No PCR inhibition was observed on retest, however the sample was also negative upon dilution (no amplication). To rule out the possibility for a potential false negative, we retested this sample with the same set of forward and reverse primers, instead using Amplitaq Mastermix instead of Taqman Environmental Mastermix, and the PCR result was positive. The amplified product was sequenced and was 100% identical to *Helicobacter* sp. GCBI1. Therefore, we report the qPCR result as inconclusive, but the conventional PCR result as positive (S1 Table).

Samples were run in singlicate on Bio-Rad Mini-Opticon Real-Time PCR detection system (Bio-Rad Laboratories, Hercules, CA, USA) under the following cycling conditions: 95˚C for 10 minutes, followed by 50 cycles of 95˚C for 15 seconds and 55˚C for 1 minute. A synthetic plasmid was created with the primer and probe binding sites to test the sensitivity and efficiency of the qPCR assay.

## Bacteriology

Samples for bacterial culture, including feces, cloacal swabs and blood from both iguana species, were collected, inoculated into *Helicobacter*-specific media [12] and shipped to the Division of Comparative Medicine at MIT where they were stored at -80˚C until analysis. Culture technique was performed as previously described for *Helicobacter* species [12].

## Results

### Mortality events and epidemiology

While canvassing the QEIIBP on May 5, 2015, the BIRP staff observed a free-ranging Grand Cayman blue iguana (Studbook No. 1894) displaying signs of hindlimb paresis and severe lethargy. The animal was taken to Island Veterinary Services and died the same day. Intra- and extracellular spiral-shaped bacteria were identified on peripheral blood smear. On May 11, 2015, another free-ranging Grand Cayman blue iguana (Studbook No. 2595), whose territory overlapped with the first case, was found with similar clinical signs. This animal was also positive for spiral-shaped bacteria on a peripheral blood smear, and recovered. These two individuals represent the first *Helicobacter* cases identified in Grand Cayman blue iguanas.

A temporally and geographically localized mortality event affecting green iguanas occurred in the West Bay and George Town regions of Grand Cayman in 2014. This event included approximately 30 individuals and spanned from April through June; antemortem clinical investigations were not performed. There was no suspicion of *Helicobacter* infection in these animals at the time of the initial investigation, and they were reexamined as potential *Helicobacter* cases after identification of the disease in Grand Cayman blue iguanas.

In total, there were 19 Grand Cayman blue iguanas, three live and 16 dead, and three green iguanas, all dead, included in this report. Nine Grand Cayman blue iguana cases from March through September 2015, nine cases from May through September 2016 and one case from 2017 met the criteria for inclusion in this report. Based on our case definition, 11 of the 19 Grand Cayman blue iguanas and two of the three green iguanas included in this report were positive for *Helicobacter* sp. GCBI1 (Table 1). All affected animals were adults; ages were known for nine of the 11 positive Grand Cayman blue iguanas and none of the green iguanas. *Helicobacter* positive Grand Cayman blue iguanas had a mean age of 16.8 years, with the youngest being 7.7 years old on presentation and the oldest 24.1 years old. Seven of the positive Grand Cayman blue iguanas were female and four were male, and both positive green iguanas were male. In 2015, the first confirmed *Helicobacter* sp. GCBI1 positive Grand Cayman blue iguana presented on May 5[th]. Another positive case presented on May 11[th] and the final positive case presented on September 29[th]. All of the *Helicobacter* positive Grand Cayman blue

**Table 1. Summary of cases included in this report.**

| Species | Studbook No. or other ID | Sex | Date of onset of clinical signs | Date of death | Helicobacter positive per case definition | Cause of death | GCBI1-specific PCR[1] | Silver stain[2] | Postmortem lesions[3] | | |
|---|---|---|---|---|---|---|---|---|---|---|---|
| | | | | | | | | | Muscle lesions | Epicarditis | Splenic lesions |
| Green iguana | N2014-0281 | M[4] | Unknown | 5/1/2014 | YES | Euthanasia; *Helicobacter* infection | POS | POS | ++ | 0 | 0 |
| Green iguana | N2014-0393 | M | 5/15/2014 | 5/15/2014 | NO | Euthanasia; inconclusive | NEG | NEG | ++ | 0 | 0 |
| Green iguana | N2014-0508 | M | 6/29/2014 | 6/29/2014 | YES | *Helicobacter* infection; gastritis and bacterial sepsis | POS | POS | ++ | 0 | 0 |
| Blue iguana | 510 | M | FD[7] | 3/29/2015 | NO | Bacterial sepsis | NEG | NEG | 0 | 0 | PMA[8] |
| Blue iguana | 2582 | F[5] | FD | 5/1/2015 | NO | Trauma (presumed predatory) | NEG | NEG | 0 | 0 | N/A[9] |
| Blue iguana | 1894 | F | 5/2/2015 | 5/5/2015 | YES | *Helicobacter* infection | NEG | POS | 0 | 0 | +++ |
| Blue iguana | 2595 | M | 5/10/2015 | SURVIVED | YES | SURVIVED | POS | N/A | N/A | N/A | N/A |
| Blue iguana | 956 | F | FD | 5/30/2015 | NO | Trauma (presumed predatory) | NEG | NEG | + | 0 | 0 |
| Blue iguana | 2924 | M | FD | 6/8/2015 | NO | Trauma | NEG | NEG | + | 0 | N/A |
| Blue iguana | 949 | M | FD | 6/21/2015 | NO | Trauma (presumed predatory) | NEG | NEG | + | 0 | ++ |
| Blue iguana | 3003 | U[6] | FD | 6/24/2015 | NO | Trauma (presumed predatory) | NEG | NEG | 0 | N/A | N/A |
| Blue iguana | 819 | F | 9/29/2015 | 9/30/2015 | YES | *Helicobacter* infection | POS | NEG | ++ | 0 | + |
| Blue iguana | 786 | M | 5/31/2016 | 6/3/2016 | YES | *Helicobacter* infection | POS | POS | ++ | ++ | ++ |
| Blue iguana | 895 | F | 5/31/2016 | 6/4/2016 | YES | *Helicobacter* infection; hemoceolom, yolk embolization | NEG | NEG | 0 | ++ | 0 |
| Blue iguana | 1046* | F | 6/1/2016 | 6/18/2016 | YES | Gastroenteritis and bacterial sepsis | NEG | NEG | ++ | 0 | 0 |
| Blue iguana | 1278 | F | FD | 6/1/2016 | YES | *Helicobacter* infection; bacterial sepsis | POS | POS | 0 | 0 | +++ |
| Blue iguana | 3054 | F | FD | 6/3/2016 | YES | *Helicobacter* infection; coelomitis, yolk embolization | POS | POS | ++ | ++ | PMA |
| Blue iguana | 784 | F | 6/10/2016 | SURVIVED | YES | SURVIVED | POS | N/A | N/A | N/A | N/A |
| Blue iguana | 2156 | M | 6/10/2016 | SURVIVED | YES | SURVIVED | POS | N/A | N/A | N/A | N/A |
| Blue iguana | 2347 | F | FD | 6/26/2016 | NO | Inconclusive | NEG | NEG | 0 | 0 | PMA |
| Blue iguana | 537** | M | 6/1/2016 | 9/25/2016 | YES | Inconclusive | NEG | NEG | + | 0 | 0 |

*(Continued)*

**Table 1.** (Continued)

| Species | Studbook No. or other ID | Sex | Date of onset of clinical signs | Date of death | Helicobacter positive per case definition | Cause of death | GCBI1-specific PCR[1] | Silver stain[2] | Postmortem lesions[3] | | |
|---|---|---|---|---|---|---|---|---|---|---|---|
| | | | | | | | | | Muscle lesions | Epicarditis | Splenic lesions |
| Blue iguana | 2054 | M | FD | 2/21/2017 | NO | Bacterial sepsis | NEG | NEG | 0 | 0 | 0 |

[1]. Includes both qPCR targeting GCBI1 and conventional PCR with sequencing to confirm GCBI1.

[2]. Positive cases do NOT include those in which silver stain identification of spirilliform bacteria was limited to the gastrointestinal tract.

[3]. See text for more detail regarding postmortem lesions.

[4]. Male.

[5]. Female.

[6]. Unknown sex.

[7]. Found dead.

[8]. Postmortem autolysis too severe for tissue interpretation.

[9]. Test not performed.

\* Blue iguana 1046 had spirilliform bacteria on blood smear at her initial presentation; she improved initially, but died after a short second illness 17 days later.

\*\* Blue iguana 537 initially presented in June 2016, was confirmed via blood smear as *Helicobacter* positive, and responded to treatment; he was found dead in September 2016, at which time infection was not identified.

iguanas in 2016 initially presented between May 31$^{st}$ and June 10$^{th}$. All positive blue iguanas were from the QEIIBP, either free-ranging on grounds or in the captive breeding facility (Fig 1). The green iguana mortality event in 2014 occurred in April through June, and all individuals were from the West Bay or George Town regions (Fig 1). Of the three green iguanas included in this study, two were euthanized due to poor prognosis associated with clinical disease and to investigate the ongoing mortality event and the third was found dead; two were from West Bay and one from George Town. Euthanasia was performed by destruction of the brainstem and decapitation in the field or by intravenous barbiturate administration.

## Climatology

The *in situ* daily records from the QEIIBP, which are complete for the years 2014–16, were utilized for comparison to documented dates of Grand Cayman blue iguana *Helicobacter* infection and associated mortality. In addition, despite being considerably wetter than the QEIIBP and the reserves on the eastern end of the island, the availability of a 43-year daily precipitation record from Owen Roberts Airport (on the western side of the island) made it possible to assess long term overall multidecadal trends in rainfall for Grand Cayman.

Daily rainfall measurements at the QEIIBP are suggestive of a link between short-term weather patterns and *Helicobacter* infections in Grand Cayman blue iguanas. Rainfall amounts at the QEIIBP and dates of recorded *Helicobacter* infections for April to October in 2015 and 2016 are presented in Fig 2. Wet season onset in both years occurred with significant rainfall events in May following multi-week spans of little if any rainfall. In 2015, two-day rainfall totaled 17 mm on 1–2 May, and in 2016 two-day rainfall of 35 mm on 29–30 May; the preceding 11 days in both years featured no observed rainfall. Ten of 11 confirmed infections occurred 1–12 days following the initial rainfall date. An outlier case on 29 Sept 2016 followed significant rainfall events five days (18 mm) and three days (12 mm) prior to the recorded date of death.

Multidecadal precipitation data revealed that the *Helicobacter* infections took place in the driest multi-year period in recent decades (Fig 3). The four-year (2014–18) drought was the first multi-year period of negative rainfall anomalies since the 1980s. Annual rainfall during

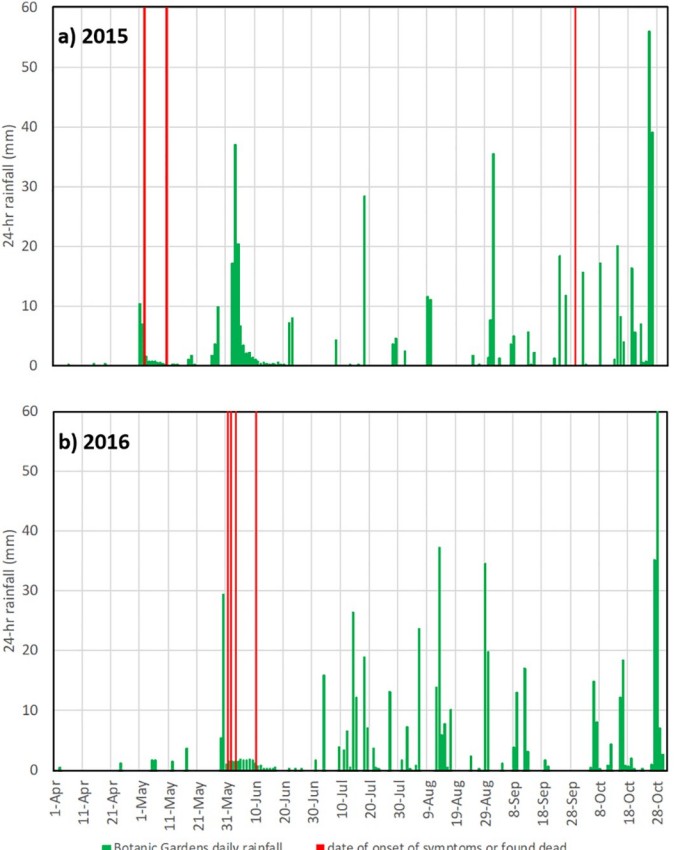

**Fig 2. Queen Elizabeth II Botanic Park daily precipitation.** Precipitation measurements in mm (green bars) for the April-October periods for a) 2015 and b) 2016. Dates of Grand Cayman blue iguana onset of signs attributed to *Helicobacter* infection or dates *Helicobacter* positive specimens were found deceased are designated by red lines. For the ease of visualization, presented are the precipitation measurements bracketing the time period when the *Helicobacter*-related mortalities occurred.

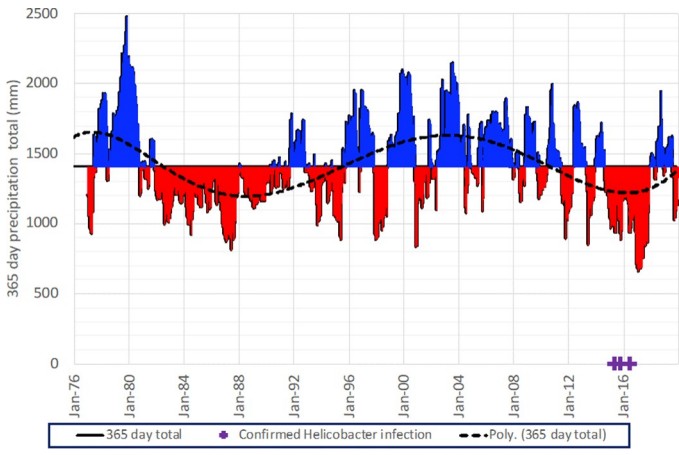

**Fig 3. Grand Cayman—Owen Roberts Airport annual and decadal precipitation anomalies 1976–2019.** Precipitation shown as 365-day totals compared to the annual mean (1,406 mm). Dashed line indicates 6th order polynomial trend showing ~28-year periodic oscillations in annual mean rainfall with an amplitude of 430 mm. Dates of Grand Cayman blue iguana confirmed *Helicobacter* infections in 2015–16 are indicated along the timeline.

2015–16, when Grand Cayman blue iguana *Helicobacter* infections were recorded, averaged approximately 200 mm below the long-term mean. We note that while Grand Cayman is occasionally affected by the passage of hurricanes and lesser intensity tropical cyclones, none of the *Helicobacter* infections occurred close in time to tropical cyclone activity.

## Clinical

Nine of the 19 Grand Cayman blue iguanas included in this report presented to Island Veterinary Services due to clinical illness. All but one (Studbook No. 819) presented within 48 hours of onset of clinical signs. Studbook No. 819 had been missing for four to seven days prior to presentation. The BIRP wardens described the animals as previously healthy until they noticed a sudden change in behavior, with animals becoming weak, inappetent, reluctant to move, and sometimes displaying abnormal gait with dragging of their hind legs. During the 2014 green iguana mortality event, affected animals were seen to be weak and some were described as appearing paralyzed, but physical examination or other antemortem diagnostics were not performed.

In the Grand Cayman blue iguanas, clinical signs at presentation included: lethargy (9 of 9); inappetence (8 of 8, with no record of the appetite status in one case); altered state of mentation—depression or stupor (7 of 9); and dehydration (5 of 8, with no record of hydration status in one case). All animals were presented in good body condition. Specific neurologic deficits, beyond depression and stupor, were not noted on physical examination in any of the animals (e.g. no proprioception deficits, ataxia, nystagmus, head tilt, etc.). Five of nine animals presented with signs of hemorrhage: three with hemorrhagic discharge from the oral cavity (n = 2, Studbook Nos. 1894 and 2595) or the cloaca (n = 1, Studbook No. 784); one animal (Studbook No. 895) developed severe mucous membrane pallor subsequent to rapidly progressing anemia resulting from hemocoelom, confirmed postmortem; and one animal (Studbook No. 786) presented with lingual petechiae.

In eight of nine cases, spiral-shaped bacteria consistent with *Helicobacter* spp. were noted during microscopic evaluation of smears, confirming active bacteremia (Fig 4). Spiral-shaped bacteria disappeared from the blood subsequent to antibiotic treatment in the three cases with follow up blood smears: two had negative smears two days after the presentation (Studbook Nos. 2595 and 895), while the third case was still positive two days after presentation, but negative nine days after presentation (Studbook No. 1046). Complete blood counts were performed in three cases with the most significant finding being marked azurophilia in all three. All other cell counts were within normal ranges (Table 2).

Biochemistry was performed on all nine iguanas on presentation, and three animals were retested once or twice as follow up, providing a total of five follow up profiles. The most prevalent changes on admission profiles included elevations in creatine kinase (CK) in five of nine animals and lactate dehydrogenase (LDH) in all three animals tested, as well as decreased glucose levels in six of nine (Table 3). Although total protein levels were overall normal, the albumin measurement using bromocresol green methodology was at or below the low end of the range in seven of nine animals at admission. Normokalemia with hyponatremia was identified in six of nine animals (Table 3). Follow up testing results were highly individual on disease progression of each case. In two cases (Studbook No. 895 died 5 days after the onset of clinical signs and Studbook No. 1046 died 17 days after the onset of clinical signs), the trends included elevations in uric acid levels, and declining total protein, albumin measurement and calcium, sodium and creatine kinase levels. In the case of Studbook No. 537 (which initially improved and was discharged, but died 3.5 months later), all initial abnormal values tended to normalize or approach normal ranges during the initial presentation: creatine kinase and uric acid declined, while total protein, albumin measurement and sodium increased (Table 3).

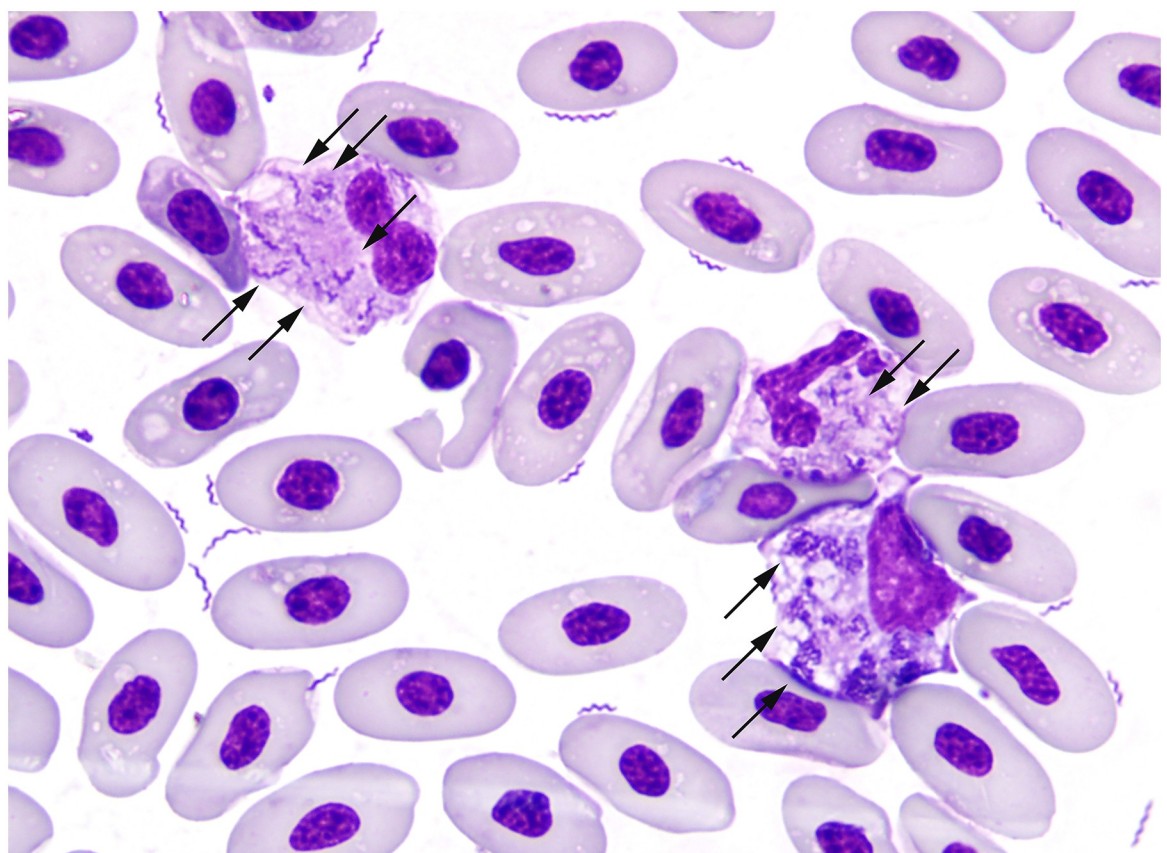

**Fig 4. Grand Cayman blue iguana peripheral blood smear revealing abundant intra- and extracellular spiral-shaped bacteria.**
Bacteria were identified as *Helicobacter* sp. GCBI1 via molecular techniques. Arrows highlight the intracellular organisms within monocyte cytoplasm. Studbook No. 786. Modified Wright-Giemsa stain ("Dip Quick"), 1000x original magnification with oil immersion.

All nine animals had the PCV measured on presentation, and three had follow up testing. Anemia was a common finding, with five of 12 samples having a PCV below the normal range (two on initial presentation and three on follow up tests), and two of 12 samples showed a marginally increased PCV of 45% (normal range 24–45%). All three follow up PCVs were lower than the values on presentation, and they originated from animals that ultimately died. Two of the three follow up PCV measurements decreased by greater than 50% over two days, which we considered too great to attribute solely to the dilution effect of fluid therapy. Both animals with marginally increased PCV survived (Table 3).

Whole body, conscious, dorso-ventral radiographs were performed at admission on all nine cases, and did not reveal any significant abnormalities. Gravidity was confirmed in four out of five females. Abdominal ultrasound was performed in one case, a gravid female suspected of concomitant coelomitis, without revealing any further diagnostic information.

Empiric antibacterial treatment was instituted in all cases except one, in which death occurred before antibiotic treatment was started (Studbook No. 1894). The treatment consisted of a combination of amikacin (5 mg/kg, Amikacin Sulfate USP, Teva, Petah Tikva, Israel) and ampicillin (20 mg/kg, Ampicillin for injection USP, Sandoz, Holzkirchen, Germany), both administered intramuscularly once daily. One case was treated using intramuscular enrofloxacin (10 mg/kg, Baytril 2.5%, Bayer AG, Leverkusen, Germany).

**Table 2. Hematology results for three *Helicobacter* sp. GCBI1 positive Grand Cayman blue iguanas.**

| Studbook number | Date | WBC count (2.4–22 $10^6$/µL) | Heterophils % | Lymphocytes % | Azurophils % | Eosinophils % | Basophils % | Absolute Heterophils 600–14500 (/µL) | Absolute Lymphocytes 40–8600 (/µL) | Absolute Azurophils 0–900 (/µL) | Eosinophils 0–1500 (/µL) | Basophils 0–1500 (/µL) | Thrombocytes |
|---|---|---|---|---|---|---|---|---|---|---|---|---|---|
| Reference intervals (units) [4] | | 2.4–22 ($10^6$/µL) | % | % | % | % | % | 600–14500 (/µL) | 40–8600 (/µL) | 0–900 (/µL) | 0–1500 (/µL) | 0–1500 (/µL) | |
| 819 | 9/29/2015 | 17.5 | 62 | 23 | 11 | 0 | 4 | 10850 | 4025 | 1925 | 0 | 700 | Adequate |
| 1894 | 5/5/2015 | 20 | 26 | 24 | 48 | n/a | 2 | 5200 | 4800 | 9600 | n/a | 400 | Adequate |
| 2595 | 5/11/2015 | 12 | 63 | 27 | 10 | 0 | 0 | 7560 | 3240 | 1200 | 0 | 0 | Adequate |

**Table 3. Biochemical assay and peripheral blood smear results for nine *Helicobacter* sp. GCBI1 positive Grand Cayman blue iguanas.**

| Studbook number | Date | Blood smear | PCV* | ALKP | ALT | AST | BA | CK | LDH | CHOL | UA | GLU | CA | PHOS | TP | ALB | GLOB | K | NA |
|---|---|---|---|---|---|---|---|---|---|---|---|---|---|---|---|---|---|---|---|
| Reference interval | | | 24–45 | 18–135 | 0–69 | 11–261 | 0.9–45.1 | 190–7000 | 22–877 | 0.9–9.2 | 23.8–565 | 7.6–17 | 2.7–6.7 | 1–5.9 | 47–101 | 22–42 | 25–59 | 0.9–7.6 | 159–192 |
| Units | | | % | U/L | U/L | U/L | µmol/L | U/L | U/L | mmol/L | µmol/L | mmol/L | mmol/L | mmol/L | g/L | g/L | g/L | mmol/L | mmol/L |
| Source | | | [4] | [4] | [4] | [4] | [4] | [15] | [4] | [4] | [4] | [4, 15] | [4] | [4] | [4, 15] | [4, 15] | [4, 15] | [4] | [4] |
| 819 | 9/29/2015 | neg[1] | 33 | 21 | 28 | 219 | n/a | 19072 | 5746 | 0.91 | 202.23 | <0.56 | 2.33 | 1.91 | 66 | 22 | 44 | 2 | 167 |
| 1894 | 5/5/2015 | pos[2] | 28 | 3 | 132 | 1548 | n/a | 12085 | 15726 | 2.36 | 481.79 | 3.55 | 5.2 | 3.23 | 66 | 22 | 44 | 4.4 | 158 |
| 2595 | 5/11/2015 | pos | 45 | 12 | 56 | 439 | n/a | 9060 | 1092 | 0.62 | 1011.16 | <0.56 | 2.8 | 4.59 | 68 | 24 | 44 | 1.4 | 169 |
| 537 | 6/1/2016 | pos | 44 | n/a | n/a | 34 | <35 | 1165 | n/a | n/a | 333.09 | 10.27 | 2.93 | 1.07 | 69 | 19 | 50 | 1.6 | 157 |
| | 6/3/2016 | n/a[3] | 22 | n/a | n/a | 401 | 5 | 8025 | n/a | n/a | 142.75 | 7.44 | 2.88 | 0.78 | 51 | 13 | 37 | 6.3 | 161 |
| | 6/10/2016 | n/a | n/a | n/a | n/a | 300 | <35 | 3202 | n/a | n/a | 285.5 | 9.6 | 3.2 | 1.74 | 67 | 17 | 50 | 3.7 | 178 |
| 786 | 6/1/2016 | pos | 43 | n/a | n/a | 1553 | 37 | >> | n/a | n/a | 832.72 | 0.22 | 2.68 | 2.49 | 59 | 11 | 48 | 5.8 | 152 |
| 895 | 6/1/2016 | pos | 26 | n/a | n/a | 21 | 0 | >> | n/a | n/a | 434.2 | 18.15 | >5 | 5.36 | 85 | 24 | 61 | 5 | 144 |
| | 6/3/2016 | neg | 3 | n/a | n/a | 103 | 0 | 1927 | n/a | n/a | 594.8 | 29.25 | >5 | 4.13 | 36 | 9 | 0 | 6.4 | 137 |
| 1046 | 6/1/2016 | pos | 17.4 | n/a | n/a | 48 | <35 | 3360 | n/a | n/a | 356.88 | 3.66 | 2.85 | 1.39 | 54 | 10 | 44 | 3.3 | 150 |
| | 6/3/2016 | pos | 16 | n/a | n/a | 187 | 135 | 3207 | n/a | n/a | 273.61 | 12.82 | 2.3 | 0.61 | 40 | <10 | >30 | 4.2 | 160 |
| | 6/10/2016 | neg | n/a | n/a | n/a | 110 | <35 | 2211 | n/a | n/a | 624.54 | 7.6 | 1.95 | 1.2 | 36 | <10 | >26 | 2.8 | 162 |
| 784 | 6/12/2016 | pos | 22 | n/a | n/a | 26 | 0 | 953 | n/a | n/a | 279.56 | 8.77 | 4.1 | 2.97 | 79 | 14 | 65 | 4.4 | 155 |
| 2156 | 6/12/2016 | pos | 47 | n/a | n/a | 141 | 18 | 5001 | n/a | n/a | 285.5 | 4.72 | 1.23 | 4.52 | 81 | 17 | 64 | 2.5 | 174 |

*Biochemical test abbreviations: PCV = packed cell volume; ALKP = alkaline phosphatase; ALT = alanine aminotransferase; AST = aspartate aminotransferase;

BA = bile acids; CK = creatine kinase; LDH = lactate dehydrogenase; CHOL = cholesterol; UA = uric acid; GLU = glucose; CA = calcium; PHOS = phosporus; TP = total protein; ALB = albumin; GLOB = globulins; K = potassium; NA = sodium.

[1]: Negative for circulating spiral-shaped bacteria.

[2]: Positive for circulating spiral-shaped bacteria.

[3]: Test not performed.

Rehydration was achieved through either intra-osseous constant rate infusion of crystalloid fluids (lactated Ringers, Hospira UK Ltd., Maidenhead, UK) at 4 to 10 ml/kg/h, depending on the severity of the case and level of dehydration, or subcutaneous boluses of 100–150 ml of fluids administered to iguanas every 12 hours. Warming lamps and electrical heat pads in the iguanas' enclosures were used for temperature control, and the animals were placed outside whenever possible for natural sunlight exposure. Anorectic but conscious animals were tube

fed using a nutritional supplement formula (Critical Care—Herbivore, Oxbow Animal Health, Omaha, NE, USA). Oral calcium gluconate (50 mg/kg) every 12 hours was used to supplement animals that were gravid or hypocalcemic on admission biochemistry.

Meloxicam (0.5 mg/kg oral once a day, Metacam®, Boehringer Ingelheim, Ingelheim am Rhein, Germany) or carprofen (4 mg/kg daily either subcutaneously or oral, Rimadyl®, Zoetis Animal Health, Parsippany, NJ, USA) was used for analgesia in seven cases.

Four out of the nine treated iguanas survived (Studbook Nos. 2595, 784, 2156 and 537) and were discharged five to nine days after presentation, and moved into a quarantine facility where they remained for up to three months before being released back into the general population (three animals were captive and one was free-ranging). Three are still alive at the time of preparation of this manuscript and one died 3.5 months after discharge (Studbook No. 537). One additional treated iguana also seemed to have survived the initial *Helicobacter* infection, but disease progressed for 17 days before death (Studbook No. 1046). Evidence of *Helicobacter* infection was not identified in these two animals at necropsy. The remaining four animals died within four days of onset of clinical signs. Overall, the recovery after the onset of treatment was very slow, with complete return to normal behavior and eating patterns occurring months after discharge in the surviving cases.

## Pathology

From 2015 to 2017, 16 Grand Cayman blue iguana mortalities meeting the inclusion criteria for this report were necropsied, including the two that survived their initial presentation, the four that were treated and did not survive, and 10 that were found dead with no preceding historical information. Additionally, three green iguana necropsies were performed during the West Bay/George Town mortality event in 2014 (Table 1).

Of the 16 Grand Cayman blue iguana necropsy cases, six were confirmed as positive for *Helicobacter* sp. GCBI1 at necropsy, and five of those were female. Two animals appeared to survive the initial infection (Studbook Nos. 1046 and 537, a female and male, respectively), as necropsy findings and ancillary diagnostics suggested that mortality was not directly related to *Helicobacter* sp. GCBI1 infection. Death of Studbook No. 1046, who died 17 days after presentation, was attributed to gastroenteritis and bacterial sepsis, with no evidence of *Helicobacter* infection at necropsy, despite being positive for circulating spiral-shaped bacteria at initial presentation (Tables 1 and 3). Studbook No. 537 was also confirmed positive via blood smear at initial presentation in June 2016, was released to the quarantine facility, and was found dead approximately 16 weeks later in September 2016; *Helicobacter* sp. GCBI1 infection was not identified at necropsy and a cause of death was not determined (Table 1).

No consistent lesions were identified via gross or histologic examination of *Helicobacter* sp. GCBI1 positive animals. The most commonly identified postmortem lesions in positive Grand Cayman blue iguanas included degeneration and/or necrosis of skeletal muscle (3 of 6), proliferation of splenic histiocytes or histiocytic splenitis (4 of 5), acute hemorrhage (3 of 6) and epicarditis (histiocytic to mixed inflammation, occasionally with a fibrinous component; 3 of 6) (Table 1, S1 Table). Soft tissue edema, thrombosis and yolk embolization were each identified in two positive Grand Cayman blue iguana necropsies. Skeletal muscle degeneration and/or necrosis (5 of 10), histiocytic splenitis (1 of 5), and hemorrhage (6 of 10) were also identified at necropsy of Grand Cayman blue iguanas that were not *Helicobacter* sp. GCBI1 positive at the time of death (Table 1, S1 Table). However, four non-*Helicobacter* positive cases with hemorrhage died as a result of traumatic events and one died of bacterial sepsis, both of which could have been responsible for the hemorrhage seen postmortem; trauma was not identified or known in any of the *Helicobacter* sp. GCBI1 positive animals (S1 Table).

Two of the three green iguanas necropsied in 2014 were positive for *Helicobacter* sp. GCBI1. Both exhibited moderate multifocal degeneration and necrosis of the skeletal muscle. Similar, less severe, lesions were also present in the one green iguana in which the organism was not identified (Table 1). No other lesion was detected in more than one individual; lesions detected in one of the two *Helicobacter* sp. GCBI1 positive green iguanas included a myocardial granuloma, thrombosis of a great vessel, and pulmonary hemorrhage (S1 Table).

Detection of spiral-shaped bacteria in tissue was accomplished via silver stain, as organisms were not visualized with routine H&E staining. The morphology of spiral-shaped bacilli in tissue ranged from subtly spiral-shaped with superficial undulations to prominent spiraling, similar to what was seen cytologically. High powered magnification (1000x with oil immersion) was required in some cases to identify the spiral morphology (Fig 5). Most tissues in which spiral-shaped bacteria were identified were otherwise normal; this was especially true in Grand Cayman blue iguanas. Even those cases which had lesions (e.g. epicarditis), spiral-shaped bacteria were often present in unaffected areas, or not identified in these tissues at all. The exceptions included both of the positive green iguanas, which had spiral-shaped bacteria in the skeletal muscle lesions and in a thrombosed great vessel, and in one Grand Cayman blue iguana that had abundant bacteria throughout lesions and healthy tissues (Studbook No. 3054). Four of the eight *Helicobacter* sp. GCBI1 positive Grand Cayman blue iguanas

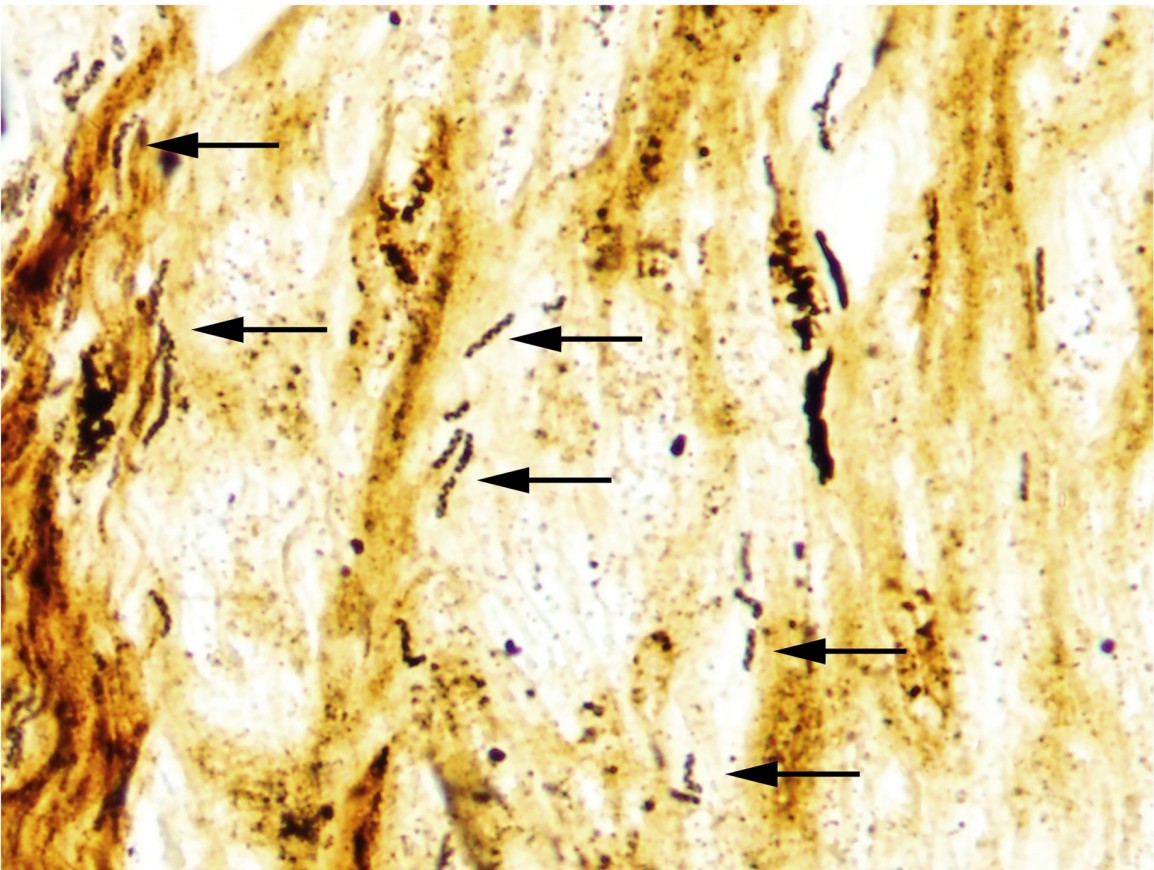

**Fig 5. Abundant argyrophilic spiral-shaped bacteria in the heart of a Grand Cayman blue iguana.** Bacteria are present in the cardiac tissue of otherwise normal heart. Arrows highlight some of the spiral-shaped organisms. Studbook No. 1278. Silver stain, original magnification 1000x with oil immersion.

necropsied had spiral-shaped bacteria in at least one non-intestinal tissue, as detected by silver staining. They were visualized in the heart of four cases and the skeletal muscle in two cases. Spiral-shaped bacteria were identified in the skeletal muscle of both *Helicobacter* sp. GCBI1 positive green iguanas; they were not present in the heart of either case (Table 1, S1 Table).

## Molecular diagnostics

To further investigate the iguana clinical cases and mortalities and identify the spiral-shaped bacteria seen on blood smears in initial cases, conventional PCR was performed in three independent laboratories using either a pan-bacteria and/or a pan-*Helicobacter* PCR assay targeting the 16S rRNA gene (S1 Table). Over the course of this investigation, blood and/or tissue from 14 total iguanas that met the inclusion criteria for this report were tested via conventional PCR. Of the 14 animals tested, six were positive for a *Helicobacter* sp. (S1 Table). BLASTN analysis of this sequence showed that the closest match in GenBank shared 95.1% identity with *Helicobacter* sp. 11S02629-2 (Taxon 2), previously identified in a Greek tortoise (*Testudo graeca*) (GenBank No. KJ081205). The Grand Cayman blue iguana *Helicobacter* sequence was only 92% identical to other reptilian *Helicobacter* spp. from *Lacertilia* (KJ081204; KJ081206; KJ081207; KJ081209; KJ081210), and 93% identical to a *Helicobacter* sp. identified in a septic pancake tortoise (*Malacochersus tornieri*, FJ667779), and was given the provisionary name, *Helicobacter* sp. GCBI1. Longer DNA sequence of the bacterial 16S rRNA gene was obtained from the initial case (Studbook No. 2595) and was used as the reference sequence. This representative sequence was deposited in GenBank under accession number MG910456. Portions (between 873 to 1422 bp) of DNA sequence were also recovered from the three laboratories, from all six positive samples (Studbook No. 2595, 819, 786, 1278, 2156, and 784), and all sequences were between 99.8%-100% identical to the reference sequence (MG910456).

We performed phylogenetic analysis on the bacterial 16S rRNA gene to determine how GCBI1 is related to other *Helicobacter* spp. As shown in Fig 6, the Grand Cayman blue iguana *Helicobacter* sp. grouped with the chelonian clade containing both pancake tortoise and Greek tortoise (*Helicobacter* Taxon 2) with an ML bootstrap value of 98.4%. *Helicobacter* Taxon 2 has been recently characterized as an enterohepatic *Helicobacter* species (EHS), whereas squamate *Helicobacter* Taxons 1, 3, 4, 6 and 7 are considered gastric *Helicobacter* species [8]. Other species grouping with *Helicobacter* sp. GCBI1 are also classified as EHS. *Helicobacter* sp. GCBI1 also shared a common ancestor with a second *Helicobacter* sp. (92% identity) cultured from fecal samples of clinically healthy Grand Cayman blue iguanas [*Helicobacter* sp. MIT 16–1353 [16] (GenBank No. NHYM00000000.1)] (Fig 6). While *Helicobacter* sp. GCBI1 was found in the blood of clinically ill animals, *Helicobacter* sp. MIT 16–1353 was only identified in feces and was not associated with septicemia.

To be able to rapidly screen for *Helicobacter* sp. GCBI1, we developed a Taqman qPCR assay specific for *Helicobacter* sp. GCBI1 to use as a screening tool. Using a linear standard curve of a 10-fold dilution series (4.5 to $4.5 \times 10^8$ copies) of the synthetic positive control, we found this assay had a slope of -3.23, an $R^2$ value of 0.996, and an efficiency of 104.2%, and consistently detected down to five copies of *Helicobacter* sp. GCBI1 per PCR reaction. We performed qPCR testing on DNA extracts (blood and tissue) from all 22 cases, including both green and blue iguanas (S1 Table). Each sample was tested by qPCR in singlicate with positive, negative, and inhibition controls. Our results show that eight of 22 animals (34.8%) were positive for *Helicobacter* sp. GCBI1 by qPCR (S1 Table). PCR inhibition was noted in one sample that was positive for a *Helicobacter* sp. by conventional PCR. This qPCR result is therefore listed in S1 Table as inconclusive (Studbook No. 819), but positive by conventional PCR.

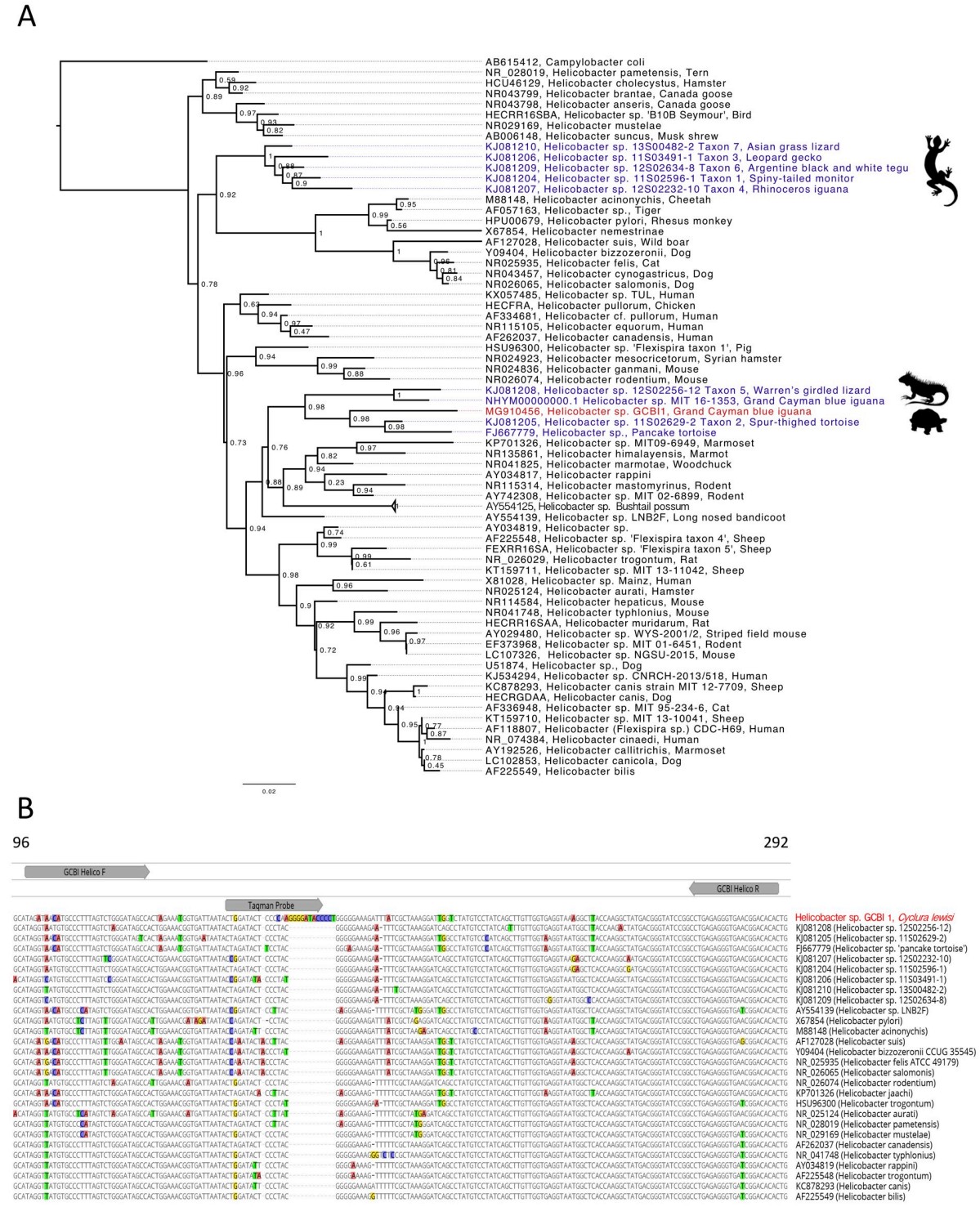

**Fig 6. Molecular analysis of *Helicobacter* sp. GCBI1.** A. Rooted phylogenetic tree of *Helicobacter* 16S rRNA nucleotide sequences, with local support values shown at the branch points. GenBank accession numbers are shown. *Helicobacter* sp. GCBI1 is shown in red. Other reptilian *Helicobacter* species are presented in blue. B. DNA sequence alignment showing primer and probe binding sites for qPCR assay targeting *Helicobacter* sp. GCBI1.

To determine if the Grand Cayman blue iguanas could be chronically harboring *Helicobacter* sp. GCBI1 in their gastrointestinal tract asymptomatically, or after recovery from infection, we collected 17 fecal and seven cloacal swab samples from 24 animals (June 26–29, 2016). Of the 24 Grand Cayman blue iguanas tested, four (Studbook Nos. 2595, 537, 2156, and 784) had been treated and survived an initial presentation with confirmed *Helicobacter* infection (16 days– 1 year prior to sample collection); others were clinically healthy and in locations either remote from or adjacent to confirmed clinical cases, and three of four animals were previously qPCR positive for *Helicobacter* sp. GCBI1. All fecal and cloacal swab samples were negative for *Helicobacter* sp. GCBI1 by qPCR (S2 Table).

Screening of free-ranging green iguanas was also performed to investigate a potential source of infection. Choanal and cloacal swabs were collected from 15 free-ranging green iguanas from the QEIIBP that were culled as part of unrelated control efforts for this invasive species in Grand Cayman (October 2019). Two of the 15 choanal swabs were positive for *Helicobacter* sp. GCBI1 by qPCR, and none of the cloacal swabs were positive. 139bp of trimmed DNA sequence was obtained from both positive samples and both sequences were 100% identical to the reference sequence (GenBank No. MG910456).

## Bacteriology

Samples collected for *Helicobacter* culture included the 17 fecal and seven cloacal swab samples from treated and healthy Grand Cayman blue iguanas described above, blood samples from six Grand Cayman blue iguanas (Studbook #786, 2156, 537, 1046, 784, and 1278) which were positive for *Helicobacter* by case definition, and 45 fecal samples from the apparently healthy culled green iguanas. None of the samples produced positive cultures for this organism, including the blood samples, of which four of the six were confirmed positive for *Helicobacter* sp. GCBI1 by PCR.

## Discussion

Our study describes a novel *Helicobacter* species, provisionally named *Helicobacter* sp. GCBI1, associated with bacteremia, illness, and/or death in a subset of endangered free-ranging and captive Grand Cayman blue iguanas in the QEIIBP. We also describe the detection of this organism associated with illness and mortality in invasive free-ranging green iguanas on Grand Cayman remote from the QEIIBP, and in apparently healthy invasive free-ranging green iguanas within the QEIIBP.

A causative relationship between *Helicobacter* sp. GCBI1 infection, illness and/or death, as defined by Koch's postulates, was not definitively attained in our study, as isolation of the organism has been unsuccessful thus far and therefore infection/challenge studies could not be pursued. From a clinical standpoint, the identification of *Helicobacter* organisms in blood smears and by PCR in sick iguanas would suggest that this organism was associated with the observed clinical signs in both iguana populations. Despite this support, the lack of disease-specific clinical signs and postmortem lesions, and the lack of consistent histopathologic identification of organisms in more common lesions, complicated confirmation of *Helicobacter* sp. GCBI1 as a distinct contributor to mortalities. The results of the thorough clinicopathologic approach to these mortality events described herein, however, reduce the likelihood of other principal causes and strengthen the probability that these mortalities are a result of infection by this novel *Helicobacter* species. Notably, neither the presence of spiral-shaped bacteria nor increased mortality levels (other than those related to known predation events) have been seen in this Grand Cayman blue iguana population in over 15 years of intensive management and monitoring. This included clinical examination and testing,

including blood smears, as well as postmortem evaluation of mortalities. The relationship between the presence of *Helicobacter* sp. GCBI1 and disease in green iguanas was more difficult to define, as bacteremia was not adequately assessed in cases without antemortem diagnostics, and this population has not been as extensively monitored as the blue iguana population. Performing necropsies on all of the animals included in this report was a fundamental step in ruling out other causes for these events, and no other consistent disease processes were identified. Toxicologic and nutritional analyses were performed on green iguana tissues during the 2014 mortality event in the West Bay and George Town regions and were unremarkable, and there was no gross or histologic evidence of toxicity or nutritional disease in the Grand Cayman blue iguanas. Finally, the response of multiple animals to antimicrobials, including clearing of organisms from circulation in the three tested iguanas, was suggestive of a bacterial cause for the illnesses and mortalities, consistent with the findings of organisms on blood smears and by PCR. While blood cultures to investigate other bacteria in circulation were not performed in sick animals (culture of blood samples was restricted to the use of *Helicobacter*-specific culture media), there was no indication of polymicrobial systemic infection in the blood smears examined.

After our initial discovery of extra- and intracellular spiral-shaped bacteria in a peripheral blood smear from a free-ranging Grand Cayman blue iguana, *Helicobacter* sp. GCBI1 was identified in a total of 11 blue iguanas in the QEIIBP from 2015–2017 (five of which survived the initial infection), two green iguana mortalities in 2014 in a separate location on Grand Cayman, and two apparently healthy free-ranging green iguanas in the QEIIBP in 2019. When present, the disease was characterized by consistent and severe, though nonspecific, clinical signs. The majority of affected Grand Cayman blue iguanas exhibited lethargy, inappetence, altered state of mentation, and dehydration. Additionally, there was bleeding or evidence of clotting abnormalities in five of nine clinical cases in this study, with hemorrhage confirmed at necropsy in the three of five that succumbed to the disease (Studbook Nos. 895, 786, and 1894). A specific cause for hemorrhage was not identified in these animals, but shock-associated pathologic responses, including disseminated intravascular coagulation (a coagulopathy often associated with sepsis), could have been contributory. Green iguanas displayed similar nonspecific clinical signs of marked lethargy and weakness, although antemortem investigation and examination in that population was limited.

Due to the small sample size and variations in methods of analysis, especially for the biochemistry values, no statistical analysis of hematology and biochemical results could be performed. As such, interpretation of the medical significance of the results was evaluated for each case individually, weighing clinical appearance and progression of clinical signs. Despite the lack of statistical analysis, some abnormal biochemistry and hematologic results, while not pathognomonic, were consistent features of *Helicobacter* sp. GCBI1 infection in the Grand Cayman blue iguana that can assist in assessing the severity of the clinical case.

High levels of CK and LDH were often noted on admission. These could partially derive from myopathy associated with capture, transport and injectable treatments, or could be directly related to infection, as muscle lesions were identified histologically in five out of eight *Helicobacter* positive Grand Cayman blue iguanas at necropsy. Cholesterol, glucose and bromocresol green albumin readings were consistently low in blue iguanas with *Helicobacter* sp. GCBI1 infection, especially on initial presentation, which was consistent with the history of inappetence. It should be noted, however, that bromocresol green does not accurately measure albumin in sauropsids or any other non-mammalian species, and these values may represent one or more different plasma proteins [17, 18]. The suspicion of an inflammatory response was also supported by the marked azurophilia noted in three cases.

Diagnostic imaging, restricted to coelomic ultrasound and non-contrast radiography, was not contributory to diagnosing *Helicobacter* sp. GCBI1 infections; however, it was helpful in determining comorbidities or establishing gravidity in females.

In the nine animals that were treated, treatment was instituted within 24–48 hours from the time any behavioral abnormalities were noticed by the park wardens, except for one case, Studbook No. 819, who had been hiding for four to seven days prior to presentation. The combination of an aminoglycoside and a penicillin was used empirically initially. This choice was based on known recommended reptile dosages for these antibiotics and their combined broad-spectrum coverage [19], and was made before *Helicobacter* was identified as the probable cause of infection. While treatment recommendations of the gastric *Helicobacter*, *H. pylori*, indicate the use of antibiotic combinations, treatment of EHS is often successful with extended use of one antibiotic. The antibiotic susceptibility profile of species associated with bacteremia in humans varies between and within species [20–24]. Isolation and sensitivity testing of *Helicobacter* sp. GCBI1 would provide better guidance on antibiotic use for future clinical cases, but bacterial isolation to date has not been successful. In addition to antibiotic treatment, supportive care was used for treatment of acute cases of *Helicobacter* sp. GCBI1 septicemia. Most animals were depressed, dehydrated, and hypoglycemic on presentation. Rapid and sustained restoration of fluid and electrolyte balance, as well as maintaining body temperature and providing appropriate nutritional support, are likely to contribute to improving success rates.

As with the clinical findings, no postmortem lesions reliably corresponded to infection in *Helicobacter* sp. GCBI1 positive iguanas. While a large percentage of positive cases had degenerative and/or necrotizing lesions in the skeletal muscle, this was also identified in non-*Helicobacter* cases. Similarly, histiocytic infiltration or inflammation of the spleen and epicarditis were repeatable findings, but were not specific to iguanas with this infection.

The only pathognomonic clinicopathologic finding was the presence of spiral-shaped bacteria on blood smears. Especially given the challenges with timely exportation of samples for PCR from these CITES-regulated species, the blood smear was an invaluable tool in diagnosis of *Helicobacter* sp. GCBI1 infection. Organisms were occasionally present in very low numbers, however, and in some cases organisms were not initially observed on cytologic examination. It was not clear if this represented cyclical bacteremia, a rapid response to treatment, or other cause.

The molecular analysis employed as part of the overall investigation into this disease was essential, from the initial identification of the spiral-shaped organism identified on blood films via pan-bacterial PCR to the creation of a qPCR assay still in use as a screening tool. One striking feature of *Helicobacter* sp. GCBI1 is a unique 14bp "mirror-like repeat" region in the 16S rRNA gene that was not found in any other *Helicobacter* species, and is the region targeted for the qPCR Taqman assay. Although this feature is uncommon in the *Helicobacter* genus, there are at least two *Helicobacter* spp. in GenBank (GenBank M88151, CLO-3; LC051629, TMU1563), as well as *Campylobacter coli* (GenBank No. AB615412), which have a unique insertion in this same region. *Helicobacter* sp. MIT 16–1353 from Grand Cayman blue iguanas does not have this 14bp insertion. Although we do not know the origin of the insertion, it may have resulted from horizontal exchange of short gene segments via other bacteria. This phenomenon has been observed in a variety of bacterial species and is thought to cause the discordance often observed when assessing the phylogenetic relationship of various target genes as compared to 16S rRNA [25].

*Helicobacter* species are common in clinically healthy animals; however, during stress, environmental change, or exposure to other diseases that compromise the host, these bacteria can cause gastritis or septicemia [9, 26]. Most affected females in this study were gravid (6 of 7), and reproductive activity in females can negatively affect their immune response, which could

have contributed to infection in these animals [27]. Specifically, gravid squamates have been shown to have decreased innate immune function as measured by bacterial lysis capacity of plasma complement [28]. Environmental stress can also negatively affect immune function [29]. The extended hydrological drought from 2014–18 evident in data presented in Fig 3 may have lowered water tables, reduced ponded rainwater availability and changed vegetation variety and availability, causing increased animal contact and conflict, as well as influencing hydration status, for Grand Cayman blue iguanas and other fauna. Humans presenting with *Helicobacter* bacteremia are often immunosuppressed or have significant underlying comorbidities. *Helicobacter cinaedi*, an EHS, is increasingly recognized as a cause of bacteremia in humans. It was first identified in HIV positive patients and has since been diagnosed in immunosuppressed patients post-transplantation, patients with hepatic cirrhosis, diabetics, and cancer patients, as well as in apparently healthy individuals [20, 23, 30].

Other identified *Helicobacter* species causing bacteremia in humans are also in the EHS clade and include *H. fennelliae* [20] and *H. pullorum* [23]. Abortion in sheep, associated with the EHS *H. bilis* and *H. trogontum*, provides additional evidence of the endovascular potential of *Helicobacter*, with hematogenous dissemination of the organism to the placenta and, subsequently, the developing fetus [31]. *Helicobacter cinaedi*, *H. fennelliae* and *H. pullorum* has each been associated with evidence of sepsis in humans, a condition with few consistent postmortem lesions. These lesions, including microthrombosis and evidence of tissue hypoxia, can be obscured by field conditions such as autolysis and freeze artefact, which were often encountered in our cases. Thrombosis was present in four *Helicobacter* positive iguanas and septic shock was presumably a significant contributor to morbidity and mortality in these cases. The two previous reports of systemic infection by spiral-shaped bacteria in reptiles offered similar postmortem findings to those identified in this case series. In the rhinoceros iguana, no lesions, other than the presence of spiral-shaped bacteria, were identified [10]. Regional cellulitis and lymphangitis and epicarditis were identified in the pancake tortoise [9].

The source of infection and the modes of transmission for *Helicobacter* sp. GCBI1 have not been identified. A contemporary introduction of the organism to the ecosystem was suspected based on its recent identification concurrent with animal mortality events, and a lack of similar morbidity or mortality identification despite health monitoring and necropsy evaluation of this population since 2001. Given that the phylogenetic analysis showed that *Helicobacter* sp. GCBI1 clustered with some *Helicobacter* species utilizing tortoise hosts, one possible source is one or more of the sympatric reptile species present on Grand Cayman, with an invasive species more likely to have introduced the organism than a species that was evolutionarily present in the ecosystem. Based on this hypothesis, choanal and cloacal swab samples were collected from free-ranging green iguanas in the QEIIBP in 2019. Two positive animals were identified via choanal swab. As these iguanas showed no evidence of clinical illness, and no green iguana mortality events have been identified in the closely monitored QEIIBP, these two individuals may be indicative of a carrier state within this species. Infection of the upper respiratory tract by *Helicobacter* spp. is not unprecedented, and has been identified in chelonians with upper respiratory disease [32]. It is therefore plausible that infection of Grand Cayman blue iguanas is, at least in part, the result of direct transmission from green iguanas, and at least one confirmed *Helicobacter* sp. GCBI1 case in a Grand Cayman blue iguana occurred after aggressive interactions with a green iguana (Studbook No. 1894). The potential for fecal-oral transmission was investigated, but not established. Popescu [33] surveyed green iguanas in the QEIIBP via fecal culture (n = 16) and PCR (n = 100) and did not identify *Helicobacter* sp. GCBI1 in any of the animals tested. And while genetic material from *Helicobacter* sp. GCBI1 was identified via PCR in frozen colon from one of the affected green iguanas from 2014 (S1 Table), this could represent circulating bacteria rather than an intraluminal/mucosal presence.

Additionally, we found no evidence for shedding of *Helicobacter* sp. GCBI1 from the cloaca or in feces of clinically healthy Grand Cayman blue iguanas adjacent to, or remote from, confirmed cases, or survivors of infection (two weeks to one year post-infection), and all cloacal swabs from the 2019 green iguana sampling were negative.

There are no clear boundaries between the captive and the free-ranging Grand Cayman blue iguana populations within QEIIBP. The free-ranging animals can come in contact with many of the captive ones through the mesh cages, and they have been known to occasionally circumvent fences to access pens. In an epidemiological context, the two groups could be considered as a single population. However, the groups have significantly different conditions, including range size (individual pens vs. unrestricted access to the QEIIBP), diet (controlled by wardens, and includes off site vegetation vs. ad libitum inside the park), and potential environmental exposures. Both free-ranging animals and captive individuals inside QEIIBP were diagnosed with *Helicobacter* infection; therefore, the type of environment (free-ranging or captive) does not seem to play a decisive role in infection and transmission. No free-ranging Grand Cayman blue iguanas outside of the QEIIBP have yet been diagnosed, indicating either a geographically restricted spreading pattern for the *Helicobacter* sp. GCBI1 or, more likely, insufficient data and challenges of close monitoring and surveillance in these other free-ranging populations.

Multiple avenues should be pursued as investigation of this disease progresses. Our current research shows an association between helicobacteremia and mortality in Grand Cayman blue iguanas, and an association between the presence of *Helicobacter* sp. GCBI1 genetic material and mortality in invasive green iguanas. The lack of specific and consistent clinicopathologic findings, however, impairs our ability to confirm the causative relationship between infection and mortality, and this is an important avenue for future research. This could include more sensitive post-mortem identification of *Helicobacter* in tissues and lesions such as *in-situ* hybridization rather than silver staining, and more comprehensive testing to better classify the negative disease state. Identification and description of a carrier state requires additional investigation. Given that we discovered the utility of choanal sampling in healthy green iguanas towards the end of our investigation, additional screening and monitoring of sympatric reptiles by multiple methods to include choanal swabs will be important to better understand the role green iguanas play in disease transmission and whether *Helicobacter* sp. GCBI1 may have emerged in one or both species from a crossover event. Such testing is recommended not only to identify potential healthy carriers, but to also determine *Helicobacter* sp. GCBI1 shedding by infected animals, treated animals before and after recovery, or during periods of physiologic or environmental stress. Testing of green iguanas outside of Grand Cayman may also contribute to our understanding of their potential role as the source of the organism and potentially identify other *Cyclura* spp. at risk of infection.

Testing for *Helicobacter* spp. environmental DNA in water sources utilized by wild reptiles may provide new insight into the epidemiology of this disease. This could be especially important in light of the climatological findings showing that mortalities appeared midway through a prolonged drought period, and within a span occurring shortly after the rains began following the wet season onset when water resources were scarce. It is possible that transmission and infection may be associated with contaminated water caused by runoff or heavy rainfall events. Research has shown that many *Helicobacter* species are waterborne and can survive prolonged periods in water (6 to > 96hr depending on water temperature and light availability) [34]. Rainfall observations associated with confirmed *Helicobacter* infection events show that in 10 of 11 cases, significant rainfall following an arid period preceded symptom onset or mortality in blue iguanas. Additionally, both reproductive behaviors and climactic conditions may have enhanced the potential for *Helicobacter* transmissions in free-ranging animals due to increased

inter- and intraspecific iguana interactions (breeding, defending territory and concentrating at remaining water sources).

Although the phylogenetic relationship of *Helicobacter* sp. GCBI1 to other *Helicobacter* spp. was assessed, further studies are needed to fully characterize this organism. Additional investigation should include growth in culture which has not yet been successful, to evaluate its antibiotic susceptibility, growth temperatures, biochemical and morphologic characteristics, analysis of additional gene sequences, and assessment of its natural behavior, including potential reservoirs. Successful culture could also allow for challenge studies to be conducted to better assess a causal relationship between infection and mortality, and a better understanding of this disease from a clinicopathological perspective. Finally, additional testing to rule out other disease processes playing a role in these mortality events should be employed. While toxicology and nutritional studies were performed on green iguana tissues in 2014 with unremarkable results (heavy metal and vitamin E analysis and GC/MS), such ancillary testing was not performed on blue iguanas.

Introduction of a novel disease agent, like *Helicobacter* sp. GCBI1, to a naïve population may have significant morbidity and mortality impacts long term. A better understanding of the epidemiology would help guide animal management decisions for the captive breeding and introduction programs for this endangered population, and potentially other managed *Cyclura* populations, to minimize the risk of future mortality events.

## Supporting information

**S1 Table. Detailed summary of cases included in this report.** *Helicobacter* sp. GCBI1 positive iguanas are shaded in grey.
(XLSX)

**S2 Table. Summary of the *Helicobacter* qPCR results of fecal and cloacal swab samples tested from Grand Cayman blue iguanas.**
(XLSX)

## Acknowledgments

The authors acknowledge and thank the staff and volunteers of the National Trust for the Cayman Islands, Blue Iguana Recovery Programme, especially the Blue Iguana Wardens; Cayman Islands Department of the Environment; St. Matthew's University School of Veterinary Medicine; Island Veterinary Services; and veterinary and histology technicians from the WCS for assistance with animal procedures and sample processing. We are grateful for Tandora Grant of the San Diego Zoo Institute for Conservation Research for technical assistance with studbook identifications. We also thank the Cayman Islands National Weather Service for providing rainfall data from Owen Roberts Airport, and Yasmin James and John Lawrus for providing rainfall data from the QEIIBP.

## Author Contributions

**Conceptualization:** Kenneth J. Conley, Tracie A. Seimon, Ioana S. Popescu, James G. Fox, Paul P. Calle.

**Formal analysis:** Anton Seimon.

**Funding acquisition:** James G. Fox, Paul P. Calle.

**Investigation:** Kenneth J. Conley, Tracie A. Seimon, Ioana S. Popescu, James F. X. Wellehan, Jr., Zeli Shen, Jane Haakonsson, Anton Seimon, Ania Tomaszewicz Brown, Veronica King, Fred Burton, Paul P. Calle.

**Methodology:** Kenneth J. Conley, Tracie A. Seimon, Ioana S. Popescu, James G. Fox, Anton Seimon, Paul P. Calle.

**Resources:** Kenneth J. Conley, Tracie A. Seimon, Ioana S. Popescu, James F. X. Wellehan, Jr., James G. Fox, Jane Haakonsson, Fred Burton, Paul P. Calle.

**Software:** Tracie A. Seimon.

**Supervision:** James G. Fox, Fred Burton, Paul P. Calle.

**Validation:** Tracie A. Seimon.

**Visualization:** Kenneth J. Conley, Tracie A. Seimon, Ioana S. Popescu, Jane Haakonsson, Anton Seimon, Fred Burton.

**Writing – original draft:** Kenneth J. Conley, Tracie A. Seimon, Ioana S. Popescu, Jane Haakonsson, Anton Seimon, Fred Burton, Paul P. Calle.

**Writing – review & editing:** Kenneth J. Conley, Tracie A. Seimon, Ioana S. Popescu, James F. X. Wellehan, Jr., James G. Fox, Zeli Shen, Jane Haakonsson, Anton Seimon, Ania Tomaszewicz Brown, Veronica King, Fred Burton, Paul P. Calle.

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
