## [Decision Letter · Decision Letter 0]

21 Oct 2020

PONE-D-20-22644

Systemic *Helicobacter* infection and associated mortalities in endangered Grand Cayman blue iguanas (*Cyclura lewisi*) and introduced green iguanas (*Iguana iguana*)

PLOS ONE

Dear Dr. Conley,

Thank you for submitting your manuscript to PLOS ONE. After careful consideration, we feel that it has merit but does not fully meet PLOS ONE’s publication criteria as it currently stands. Therefore, we invite you to submit a revised version of the manuscript that addresses the points raised during the review process.

Though the manuscript has certainly merit, it will need major revisions. I agree with the statement that currently, no pathology can be associated with the presence of this bacteria in Iguana's. This has been stated by the two reviewers. It should be made very clear that currently the role of Helicobacter in the disease is unknown. In the discussion it should be explained why this cannot be concluded and what is necessary to be able to come to a conclusion on the question whether this Helicobacter species is pathogenic for Iguana's. There are also several methodological questions that will need to be addressed. Reply also to all other remarks.

We look forward to receiving your revised manuscript.

Kind regards,

Patrick Butaye, DVM, PhD

Academic Editor

PLOS ONE

Journal Requirements:

2. In your Methods section, please state the volume of the blood samples collected for use in your study.

3. In your Methods section, please include a comment about the state of the animals following this research. Were they released, euthanized or housed for use in further research? If any animals were sacrificed by the authors, please include the method of euthanasia and describe any efforts that were undertaken to reduce animal suffering.

4. In your Methods section, please provide additional location information of the study area, including geographic coordinates for the data set if available.

5.  We note that Figure 1 in your submission contain map/satellite images which may be copyrighted. All PLOS content is published under the Creative Commons Attribution License (CC BY 4.0), which means that the manuscript, images, and Supporting Information files will be freely available online, and any third party is permitted to access, download, copy, distribute, and use these materials in any way, even commercially, with proper attribution. For these reasons, we cannot publish previously copyrighted maps or satellite images created using proprietary data, such as Google software (Google Maps, Street View, and Earth). For more information, see our copyright guidelines: http://journals.plos.org/plosone/s/licenses-and-copyright.

5.1. You may seek permission from the original copyright holder of Figure 1 to publish the content specifically under the CC BY 4.0 license. 

5.2. If you are unable to obtain permission from the original copyright holder to publish these figures under the CC BY 4.0 license or if the copyright holder’s requirements are incompatible with the CC BY 4.0 license, please either i) remove the figure or ii) supply a replacement figure that complies with the CC BY 4.0 license. Please check copyright information on all replacement figures and update the figure caption with source information. If applicable, please specify in the figure caption text when a figure is similar but not identical to the original image and is therefore for illustrative purposes only.

6. Thank you for stating the following in the Competing Interests section:

We note that one or more of the authors are employed by a commercial company: Newmarket Vets4Pets.

6.1. Please provide an amended Funding Statement declaring this commercial affiliation, as well as a statement regarding the Role of Funders in your study. If the funding organization did not play a role in the study design, data collection and analysis, decision to publish, or preparation of the manuscript and only provided financial support in the form of authors' salaries and/or research materials, please review your statements relating to the author contributions, and ensure you have specifically and accurately indicated the role(s) that these authors had in your study. You can update author roles in the Author Contributions section of the online submission form.

6.2. Please also provide an updated Competing Interests Statement declaring this commercial affiliation along with any other relevant declarations relating to employment, consultancy, patents, products in development, or marketed products, etc.  

Please respond by return email with an updated Funding Statement and Competing Interests Statement and we will change the online submission form on your behalf.

Reviewers' comments:

Reviewer's Responses to Questions

**Comments to the Author**

1. Is the manuscript technically sound, and do the data support the conclusions?

Reviewer #1: No

Reviewer #2: Yes

2. Has the statistical analysis been performed appropriately and rigorously? 

Reviewer #1: N/A

Reviewer #2: I Don't Know

3. Have the authors made all data underlying the findings in their manuscript fully available?

Reviewer #1: Yes

Reviewer #2: Yes

4. Is the manuscript presented in an intelligible fashion and written in standard English?

Reviewer #1: Yes

Reviewer #2: Yes

5. Review Comments to the Author

Reviewer #1: The manuscript "Systemic Helicobacter infection and associated mortalities in endangered Grand Cayman blue iguanas (Cyclura lewisi) and introduced green iguanas (Iguana iguana)" by Conley and colleagues is a very well written paper that describes the occurence of a novel Helicobacter sp. infection in a captive group of endemic iguanas on Grand Cayman as well as in syntopically and more remotely occuring green iguanas. The story is of great interest for the readership of PLoS One both from a veterinary and a conservation perspective. However, the study has some flaws that need to be addressed prior to publication.

Although I can imagine the authors are correct with their assumptions, the biggest isssue, however, is that Koch's postulates have not been fulfilled in the present study. Therefore, the etiological role of the proposed Helicobacter is highly speculative. This should be made (even) more clear! Unfortunately, authors also missed to present any results from parasitology, bacteriology and virology, so one cannot conclude from the presented data that the Helicobacter is the sole microorganism involved. The case series has exhaustively been done using very good molecular results but failed in different approaches to isolate respective strains (attempts done only from 6 blood samples with uncertain methodology).

Here are some comments for the authors in order to improve their study (I have also uploaded a pdf version with minor remarks):

- is there an explanation why authors have not attempted to isolate the organism also from tissues?

- the etiological role is further complicated by almost identical histological findings in the Helicobacter negative group, why did authors go not further into detail with their established highly sensitive qPCR protocol? Wouldn't it be possible to investigate all tissues separately in order to better estimate the bacterial load per organ?

- how can authors explain the quite long times (18 days and 3.5 months) until death despite antimicrobial therapy in two individuals? How can the relapse in #2156 be explained?

- why did authors miss to conduct also neurological examinations in sick animals (and also present data from neural tissue)?

- despite presenting highly sophisticated wheather data I do not understand their relevance: why is precipitation data not presented for the whole year? Why should animals suffer from disease especially after raining (aggregation at water sites should not be necessary then, green iguanas remained healthy despite carrying the microorganism, 3-year drought period seems to be a higher stressor but without signs of disease)

- authors present data from 2014-2016: what happened thereafter (further losses?)?

- authors have conducted some monitoring in apparently only one additional species (green iguana) with very few to no further evidence, but sampling was a little curious (fecal samples (although choanal samples were tested positive before). Why not e.g. gastric or cloacal wahshings? Why not choanal samples in the Blue Iguanas instead? Why not sampling outside QEIIBP?

- can authors exclude a non-infectious etiology (e.g. (plant) poisoning)? Please discuss.

- are the authors really sure that their Helicobacter is really intracellular? I cannot conclude this from Fig. 4, this might be an overlay phenomenon?

- why did authors not try to find the Helicobacter OTU also in iguana food (PMID: 29457310), arthropods (PMID: 15499059) or further (iguanid) reptiles (cf. recent introduction report on Devriesea in anoles and I. delicatissima (PMID: 28970515))?

- concerning therapy: why not focusing on research in human EHS therapy instead of reptile formulae: PMID: 24274445 suggests macrolides and points out resistance in fluorchinolons

Reviewer #2: This manuscript describes the characterization of a novel Helicobacter infection and its associated mortalities in endangered iguanas. The manuscript shows interesting findings but lacks however some clarity and additional results. Below an overview of my comments:

* it is difficult to extract from the text how many alive and dead animals were included in this study and to which species they belonged. This should be clarified in the text and referred to the tables.

*Was blood only collected from the live animals? How was Helicobacter positivity investigated in the dead animals?

*Also the tables are difficult to read. What is meant with non-Helicobacter sepsis? Sepsis caused in the absence of Helicobacter species or caused by non-Helicobacter pylori Helicobacter species? Does the studbook number refers to the animal number? In table 2, I would also include the animal species.

*In general, Figures are of bad quality. I would recommend to upload figures with higher quality.

*Figure 6: phylogentic tree: why not including gastric Helicbacter species ? Does the novel Helicbacter species belongs to gastric or enterohepatic species? And are the authors convinced that this Helicobacter infection is caused by one species. Or do they think that multiple species can be involved?

*Did the authors also collect environmental samples to investigate the origin of this novel Helicbacter species since it was shown by the authors that healthy iguanas are not colonized with this species in the gut?

*Did the authors try to isolate the novel Helicobacter species from blood cultures? This would be of added value to further characterise this novel Helicobacter species. Does this species produces urease since it is known that some enterohepatic species have the ability to produce urease?

*page 22 (bottom): one animal, that survived the infection, showed a year after the same clinical signs but was now negative for Helicobacter. What was the cause then? Does this questions the pathogenic role of this Helicobacter species in iguanas?

*How would you explain the fact that Helicobacter GCBI1 was only found in blood smears but not in affected organs upon pathological examination? Would this again questions the pathogenic role in this animal species?

6. PLOS authors have the option to publish the peer review history of their article (what does this mean?). If published, this will include your full peer review and any attached files.

Reviewer #1: No

Reviewer #2: No

---

## [Author Response · Author response to Decision Letter 0]

14 Jan 2021

Editor comments and Journal requirements

- My apologies for not meeting requirements. All identified deviations have been corrected for our response and all re-submitted files are named according to the requirements.

2. In your Methods section, please state the volume of the blood samples collected for use in your study. 

- Our manuscript relied on review of medical records written at the time the animals were examined. Unfortunately, records did not include the volume of blood collected for diagnostics, but we did add to the manuscript a general description of the volumes expected. The following sentence has been added beginning at line 171: “The volume of blood collected was not specified in the reviewed medical records; generally, the sample volume depends on the diagnostic testing expected to be performed, but does not exceed 2% of the blood volume.” 

3. In your Methods section, please include a comment about the state of the animals following this research. Were they released, euthanized or housed for use in further research? If any animals were sacrificed by the authors, please include the method of euthanasia and describe any efforts that were undertaken to reduce animal suffering.

- Please note that this manuscript describes a largely retrospective study of animals that developed illness that required veterinary care in either a head-starting and captive breeding program or a free-ranging population, rather than a study based on research-purposed animals. Therefore, the available ancillary diagnostics performed and samples collected were specific for each case and are not uniform. We have added verbiage to the introduction to further clarify this point (Lines 100-103, “Through retrospective and contemporaneous investigation into the resulting and ongoing mortality events in the endangered Grand Cayman blue iguanas, here we describe the clinical, pathological, and molecular presentation of a novel Helicobacter species and associated disease.”).

- We have added the following sentence to the lines 126-129 in the ethics statement in the methods: “The welfare of animals included in this study was considered throughout their care, with analgesics used as deemed necessary. Animals that survived infection were released back into the environment where they were found, either in the captive setting or free-ranging within the QEIIBP. No animals were housed for continued research purposes”. 

- This point was further described in more detail in the results section, Lines 499-502 rather than methods due to the retrospective and record-based nature of the manuscript: “Four out of the nine treated iguanas survived (Studbook Nos. 2156, 784, 537 and 2595) and were discharged five to nine days after presentation, and moved into a quarantine facility where they remained for up to three months before being released back into the general population (three animals were captive and one was free-ranging).”

- The only euthanized animals were two of the three green iguanas; this has been added to the results as well (beginning at Line 340): “Of the three green iguanas included in this study, two were euthanized due to poor prognosis associated with clinical disease and to investigate the ongoing mortality event and the third was found dead; two were from West Bay and one from George Town. Euthanasia was performed by destruction of the brainstem and decapitation in the field or by intravenous barbiturate administration.”

4. In your Methods section, please provide additional location information of the study area, including geographic coordinates for the data set if available.

- Specific geographic coordinates were intentionally omitted to reduce unintended human interactions with this critically endangered population.

5. We note that Figure 1 in your submission contain map/satellite images which may be copyrighted. All PLOS content is published under the Creative Commons Attribution License (CC BY 4.0), which means that the manuscript, images, and Supporting Information files will be freely available online, and any third party is permitted to access, download, copy, distribute, and use these materials in any way, even commercially, with proper attribution. For these reasons, we cannot publish previously copyrighted maps or satellite images created using proprietary data, such as Google software (Google Maps, Street View, and Earth). For more information, see our copyright guidelines: http://journals.plos.org/plosone/s/licenses-and-copyright.

5.1. You may seek permission from the original copyright holder of Figure 1 to publish the content specifically under the CC BY 4.0 license. 

We recommend that you contact the original copyright holder with the Content Permission Form (https://protect-us.mimecast.com/s/GQm8Crk5A4Cw6poXUQhiEp) and the following text:

“I request permission for the open-access journal PLOS ONE to publish XXX under the Creative Commons Attribution License (CCAL) CC BY 4.0 (https://protect-us.mimecast.com/s/FKTeCv25G4fO4GKoUwPuPB). Please be aware that this license allows unrestricted use and distribution, even commercially, by third parties. Please reply and provide explicit written permission to publish XXX under a CC BY license and complete the attached form.”

5.2. If you are unable to obtain permission from the original copyright holder to publish these figures under the CC BY 4.0 license or if the copyright holder’s requirements are incompatible with the CC BY 4.0 license, please either i) remove the figure or ii) supply a replacement figure that complies with the CC BY 4.0 license. Please check copyright information on all replacement figures and update the figure caption with source information. If applicable, please specify in the figure caption text when a figure is similar but not identical to the original image and is therefore for illustrative purposes only.

 USGS National Map Viewer (public domain): https://protect-us.mimecast.com/s/Js-YCwp5J4S0P79XSl8c6I

The Gateway to Astronaut Photography of Earth (public domain): https://protect-us.mimecast.com/s/AlFhCxk5K4CmBywKixvoSv

Maps at the CIA (public domain): https://protect-us.mimecast.com/s/kqNBCyP5Lgf6JgkEsgNJbp and https://protect-us.mimecast.com/s/yKwxCzp5MjS8n39vhxwVox

NASA Earth Observatory (public domain): https://protect-us.mimecast.com/s/V9A2CADo3kirVg2MHn3g-V

USGS EROS (Earth Resources Observatory and Science (EROS) Center) (public domain): https://protect-us.mimecast.com/s/B0mTCDkr3nCjM10QFEnfVh

Natural Earth (public domain): https://protect-us.mimecast.com/s/vIX_CERv3oC1nmMKCoDe2y

- These maps were created by, and are owned by the Cayman Islands Department of the Environment. They were created using ArcGIS and Quickbird satellite imagery. The following has been added to the figure caption (beginning Line 116):

o Created by the Department of Environment, Cayman Islands Government. 2006 Quickbird satellite imagery courtesy of the Department of Environment, Cayman Islands Government. 

- Written permission to use the image has been provided from the Department of Environment and submitted with this revision

6. Thank you for stating the following in the Competing Interests section:

We note that one or more of the authors are employed by a commercial company: Newmarket Vets4Pets.

6.1. Please provide an amended Funding Statement declaring this commercial affiliation, as well as a statement regarding the Role of Funders in your study. If the funding organization did not play a role in the study design, data collection and analysis, decision to publish, or preparation of the manuscript and only provided financial support in the form of authors' salaries and/or research materials, please review your statements relating to the author contributions, and ensure you have specifically and accurately indicated the role(s) that these authors had in your study. You can update author roles in the Author Contributions section of the online submission form.

- The Funding Statement now includes commercial employment of one author (ISP) providing salary support only; this statement was supplied via reply email:

o Funding for this project was provided by the Wildlife Conservation Society’s Zoological Health Program, the Cayman Islands Government, the National Trust for the Cayman Islands, the Derald H Ruttenberg Memorial Fund (PPC), and the following grants (JGF): P30-ES0002109, R01-OD011141, and T32-OD010978. One author (ISP) was commercially employed through the duration of the study and manuscript preparation (Island Veterinary Services and Newmarket Vets4Pets); funding was limited to salary. None of the funders had any role in study design, data collection and analysis, decision to publish, or preparation of the manuscript. The specific roles of all authors are articulated in the “author contributions” section.

- A revised Author Contributions statement has also been updated on the online submission form.

6.2. Please also provide an updated Competing Interests Statement declaring this commercial affiliation along with any other relevant declarations relating to employment, consultancy, patents, products in development, or marketed products, etc. 

Within your Competing Interests Statement, please confirm that this commercial affiliation does not alter your adherence to all PLOS ONE policies on sharing data and materials by including the following statement: "This does not alter our adherence to PLOS ONE policies on sharing data and materials.” (as detailed online in our guide for authors https://protect-us.mimecast.com/s/AGuOCG6x3qtWqmGofNxJkq). If this adherence statement is not accurate and there are restrictions on sharing of data and/or materials, please state these. Please note that we cannot proceed with consideration of your article until this information has been declared.

Please respond by return email with an updated Funding Statement and Competing Interests Statement and we will change the online submission form on your behalf.

Please know it is PLOS ONE policy for corresponding authors to declare, on behalf of all authors, all potential competing interests for the purposes of transparency. PLOS defines a competing interest as anything that interferes with, or could reasonably be perceived as interfering with, the full and objective presentation, peer review, editorial decision-making, or publication of research or non-research articles submitted to one of the journals. Competing interests can be financial or non-financial, professional, or personal. Competing interests can arise in relationship to an organization or another person. Please follow this link to our website for more details on competing interests: https://protect-us.mimecast.com/s/AGuOCG6x3qtWqmGofNxJkq

- A new competing interests statement has been written and supplied via email reply:

o I have read the journal's policy and the authors of this manuscript have the following competing interests: One author (ISP) was commercially employed through the duration of the study and manuscript preparation (Island Veterinary Services and Newmarket Vets4Pets); funding from these businesses was limited to salary support. This does not alter our adherence to PLOS ONE policies on sharing data and materials.

Reviewer #1: The manuscript "Systemic Helicobacter infection and associated mortalities in endangered Grand Cayman blue iguanas (Cyclura lewisi) and introduced green iguanas (Iguana iguana)" by Conley and colleagues is a very well written paper that describes the occurence of a novel Helicobacter sp. infection in a captive group of endemic iguanas on Grand Cayman as well as in syntopically and more remotely occuring green iguanas. The story is of great interest for the readership of PLoS One both from a veterinary and a conservation perspective. However, the study has some flaws that need to be addressed prior to publication.

Although I can imagine the authors are correct with their assumptions, the biggest isssue, however, is that Koch's postulates have not been fulfilled in the present study. Therefore, the etiological role of the proposed Helicobacter is highly speculative. This should be made (even) more clear! Unfortunately, authors also missed to present any results from parasitology, bacteriology and virology, so one cannot conclude from the presented data that the Helicobacter is the sole microorganism involved. The case series has exhaustively been done using very good molecular results but failed in different approaches to isolate respective strains (attempts done only from 6 blood samples with uncertain methodology).

- We agree that definitive causation is an issue for this investigation and have tried to increase the acknowledgement of uncertainty in the manuscript. Specifically, we added the following paragraph near the beginning of the discussion (beginning Line 667): 

o “A causative relationship between Helicobacter sp. GCBI1 infection, illness and/or death, as defined by Koch’s postulates, was not definitively attained in our study, as isolation of the organism has been unsuccessful thus far and therefore infection/challenge studies could not be pursued. From a clinical standpoint, the identification of Helicobacter organisms in blood smears and by PCR in sick iguanas would suggest that this organism was associated with the observed clinical signs in both iguana populations. Despite this support, the lack of disease-specific clinical signs and postmortem lesions, and the lack of consistent histopathologic identification of organisms in more common lesions, complicated confirmation of Helicobacter sp. GCBI1 as a distinct contributor to mortalities. The results of the thorough clinicopathologic approach to these mortality events described herein, however, reduce the likelihood of other principal causes and strengthen the probability that these mortalities are a result of infection by this novel Helicobacter species. Notably, neither the presence of spiral-shaped bacteria nor increased mortality levels (other than those related to known predation events) have been seen in this Grand Cayman blue iguana population in over 15 years of intensive management and monitoring. This included clinical examination and testing, including blood smears, as well as postmortem evaluation of mortalities. The relationship between the presence of Helicobacter sp. GCBI1 and disease in green iguanas was more difficult to define, as bacteremia was not adequately assessed in cases without antemortem diagnostics, and this population has not been as extensively monitored as the blue iguana population. Performing necropsies on all of the animals included in this report was a fundamental step in ruling out other causes for these events, and no other consistent disease processes were identified. Toxicologic and nutritional analyses were performed on green iguana tissues during the 2014 mortality event in the West Bay region and were unremarkable, and there was no gross or histologic evidence of toxicity or nutritional disease in the Grand Cayman blue iguanas. Finally, the response of multiple animals to antimicrobials, including clearing of organisms from circulation in the three tested iguanas, was suggestive of a bacterial cause for the illnesses and mortalities, consistent with the findings of organisms on blood smears and by PCR. While blood cultures to investigate other bacteria in circulation were not performed in sick animals (culture of blood samples was restricted to the use of Helicobacter-specific culture media), there was no indication of polymicrobial systemic infection in the blood smears examined.”

- Additionally, the following was included within a second new paragraph towards the end of the discussion (beginning Line 884): “The lack of specific and consistent clinicopathologic findings, however, impairs our ability to confirm the causative relationship between infection and mortality, and this is an important avenue for future research.”

Here are some comments for the authors in order to improve their study (I have also uploaded a pdf version with minor remarks):

- is there an explanation why authors have not attempted to isolate the organism also from tissues?

- Many of the diagnostics are not available on-island and thus require exportation to the US (following importation of media, etc. in anticipation of mortalities). We therefore needed to prioritize what samples were pursued for additional testing. In doing so, and considering what was known about the disease in iguanas and about Helicobacter species in general, we elected to save our Helicobacter-specific culture media for samples with perceived higher likelihood of success, such as cloacal swabs, feces and peripheral blood from suspected sick animals. Necropsies were performed mostly on frozen carcasses, and long after death. Our experience (JGF) is that successful isolation of Helicobacter species from such samples is unlikely. See lines 291-294 regarding attempted culture: “Samples for bacterial culture, including feces, cloacal swabs and blood from both iguana species, were collected, inoculated into Helicobacter-specific media [12] and shipped to the Division of Comparative Medicine at MIT where they were stored at -80C until analysis. Culture technique was performed as previously described for Helicobacter species [12].” 

- We also added lines 652-658: “Samples collected for Helicobacter culture included the 17 fecal and seven cloacal swab samples from treated and healthy Grand Cayman blue iguanas described above, blood samples from six Grand Cayman blue iguanas (Studbook #786, 2156, 537, 1046, 784, and 1278) which were positive for Helicobacter by case definition, and 45 fecal samples from the apparently healthy culled green iguanas. None of the samples produced positive cultures for this organism, including the blood samples, of which four of the six were confirmed positive for Helicobacter GCBI1 by PCR.”

- the etiological role is further complicated by almost identical histological findings in the Helicobacter negative group, why did authors go not further into detail with their established highly sensitive qPCR protocol? Wouldn't it be possible to investigate all tissues separately in order to better estimate the bacterial load per organ?

- Yes, the qPCR assay targeting H. sp. GCBI-1 could quantify bacterial presence per organ for comparison, but there were a number of inter-sample variables that could not be accounted for, rendering such a comparison invalid. Variables were largely a factor of the retrospective nature of this project. As sample collection was not uniform between cases, especially in regards to frozen tissue collection, formalin fixed paraffin embedded tissue was often used instead. This could increase the opportunity for contamination in addition to reducing available DNA due to prolonged formalin immersion (exportation from Grand Cayman often resulted in prolonged formalin immersion). Additionally, the potential for cross-organ (intra-organismal) contamination at the time of collection could not be definitively excluded.

- how can authors explain the quite long times (18 days and 3.5 months) until death despite antimicrobial therapy in two individuals? How can the relapse in #2156 be explained?

- Studbook #1046 presented on 1Jun16, was treated and eventually died on 18Jun20. Spiral bacteria were present on blood smears on the 1st and 3rd of June, but not on the 10th and no evidence of ongoing Helicobacter infection was identified postmortem (silver stains and molecular testing were negative). Necropsy revealed gastroenteritis and bacterial sepsis. In this case, we suspect that Helicobacter infection was resolved, but infection, hospitalization, etc., may have predisposed this animal to opportunistic (non-Helicobacter) bacterial infection. These details were provided in the Pathology results, lines 527-530: “Death of Studbook No. 1046, who died 17 days after presentation, was attributed to gastroenteritis and bacterial sepsis, with no evidence of Helicobacter infection at necropsy, despite being positive for circulating spiral-shaped bacteria at initial presentation (Tables 1 and 3).”

- Studbook #537 initially presented on 1Jun16, and had spiral bacteria on a blood smear at that time. He improved with treatment, but was found dead on 25Sep16. Necropsy did not reveal a cause of death and all Helicobacter diagnostics were negative at that time. We therefore suspected that he was Helicobacter negative at the time of death. These details were provided in the Pathology results, along with above, lines 530-534: “Studbook No. 537 was also confirmed positive via blood smear at initial presentation in June 2016, was released to the quarantine facility, and was found dead approximately 16 weeks later in September 2016; Helicobacter sp. GCBI1 infection was not identified at necropsy and a cause of death was not determined (Table 1).”

- I will also take this opportunity to note that we adjusted our manuscript to reflect that these two individuals were not Helicobacter positive at the time of death. This included adjusting the paragraph quoted above, the introductory numbers (lines 517-534) and lesion prevalence (lines 535-550) in the pathology results section, and in other locations (lines 42-43, lines 499-511 and line 700)

- Studbook #2156 initially presented on 12Jun2016 and was positive for Helicobacter via blood smear and blood PCR at that time. She recovered and then re-presented with similar signs, although diagnostics did not reveal Helicobacter infection. Unfortunately, the only Helicobacter testing performed at that time was a blood smear, which was read as negative. As the clinical signs are not specific for Helicobacter infection, it is possible that this was an unrelated illness and I apologize for any suggestion that this was a relapse. As such, and since this re-presentation would not have qualified for inclusion in this manuscript based on our parameters, we have removed the following sentence from the manuscript (lines 511-514): “One of the surviving animals (Studbook No. 2156) presented one year after the initial episode, with the same clinical signs as a year before. Both hospitalization and recovery times were shorter, with two days of hospitalization in 2017 vs eight days in 2016. However, infection with Helicobacter sp. GCBI1 was not confirmed in 2017”. 

- why did authors miss to conduct also neurological examinations in sick animals (and also present data from neural tissue)?

- Thank you for pointing this out. Physical examinations were performed on all blue iguanas presenting clinically ill, which included assessment of the neurologic status, but we failed to specifically mention that in the manuscript. We have added the following comment (lines 415-417): “Specific neurologic deficits, beyond depression and stupor, were not noted on physical examination in any of the animals (e.g. no proprioception deficits, ataxia, nystagmus, head tilt, etc.).”

- In almost all cases, the central nervous system was not collected postmortem. In those cases in which brain was available for examination, it was unremarkable. We have added a sentence regarding tissue collection to the methods (lines 205-209): “Collected tissue sets were inconsistent, but nearly all cases included at least liver, lung, kidney, stomach, intestine and skeletal muscle in formalin (one case, Studbook No. 3003 was markedly autolyzed and only skeletal muscle and skin were collected); central nervous system tissues were infrequently included.”

- despite presenting highly sophisticated wheather data I do not understand their relevance: why is precipitation data not presented for the whole year? Why should animals suffer from disease especially after raining (aggregation at water sites should not be necessary then, green iguanas remained healthy despite carrying the microorganism, 3-year drought period seems to be a higher stressor but without signs of disease)

- Thank you for pointing this out. To clarify, we analyzed multi-decadal trends in precipitation patterns to get an understanding of the decadal above and below average variability between 1976 and 2019, which indicated that Grand Cayman was undergoing a cyclic period of drought in 2016. We then looked at the daily precipitation records for the time period between 2014-2016, not just the those few months we presented in Figure 2. For the ease of visualization, we focused on and only presented the time periods bracketing when the Helicobacter-related mortalities occurred, so we could show the reader the pattern of rainfall related to illness, and place within the multidecadal context. A comment to this effect has been added to the caption for Fig2 (lines 380-384) for clarity: “Dates of Grand Cayman blue iguana onset of signs attributed to Helicobacter infection or dates Helicobacter positive specimens were found deceased are designated by red lines. For the ease of visualization, presented are the precipitation measurements bracketing the time period when the Helicobacter-related mortalities occurred.”

- We feel it is important to document that this mortality event and illnesses occurred within a cyclical pattern of drought on Grand Cayman and that the mortalities occurred during the driest multi-year period in recent decades (fig 3 and lines 397-400). This could result in an increased potential for transmission between free-ranging individuals at reduced water sources, where animals congregate to utilize fewer water resources. Regarding infection post-rain events, we discuss the potential for runoff or heavy rainfall potentially resulting in water contamination and contributing to infection (lines 907-911). We agree that this data does not prove causation, but we feel this data is important to have already documented for future research to see if a correlation or pattern holds up during any future outbreaks on the island, especially given the research showing waterborne Helicobacter species.

- authors present data from 2014-2016: what happened thereafter (further losses?)?

- Data presented runs through 2017, but no positive cases were identified that year. Two additional Helicobacter mortalities were identified in Blue Iguanas in 2019, but results from those cases were not available when this manuscript was compiled. 

- authors have conducted some monitoring in apparently only one additional species (green iguana) with very few to no further evidence, but sampling was a little curious (fecal samples (although choanal samples were tested positive before). Why not e.g. gastric or cloacal wahshings? Why not choanal samples in the Blue Iguanas instead? Why not sampling outside QEIIBP?

- Sampling techniques have evolved throughout this event as we learn more about the disease and about helicos in non-mammalian species. Choanal swabbing of the green iguanas was done opportunistically on culled animals, based on recent reports of helicos being found in nasal swabs of chelonians, and we focused on the QEIIBP as that is where the majority of Helicobacter cases have been identified. Fecal testing had been done earlier on green iguanas based on more typical Helicobacter biology and association with the gut. Moving forward in future studies, choanal swabbing will be performed more broadly on blue iguanas and expanded to other sympatric species, including additional green iguanas.

- With regard to sampling outside of QEIIBP, all sampling to date has been done in sick blue iguanas (diagnostic) and in the blue iguanas held in the captive breeding facility (screening). The only other known populations of blue iguanas are in protected areas which are difficult to access for monitoring these populations. Sampling of healthy blue iguanas in these populations would require significant efforts in trapping and was not deemed worthy of the potential risks to these endangered animals. We have monitored for wild blue iguana mortalities from around the reserves (i.e. road fatalities), but assume that the vast majority of mortalities outside these monitored areas are never found. 

- can authors exclude a non-infectious etiology (e.g. (plant) poisoning)? Please discuss.

- We cannot definitively exclude this possibility. That being said, classic lesions associated with toxicity (e.g. hepatic or renal damage) were not seen histologically and unusual gastric content or other indication of toxicity, was not found grossly; this includes a lack of oleander plant material or evidence of heart failure that would be expected with oleander toxicity. While they most surely exist, we are not aware of other plants with potential toxicity for iguanas on Grand Cayman, and there have been few reports of toxin-related mortality events in free-ranging reptiles in the literature. The potential for chemical intoxication was discussed with the Mosquito Research and Control Unit in Grand Cayman and there was no temporal association with pesticide spraying and mortalities. Toxicity was not specifically investigated in the blue iguanas, but was pursued during the green iguana mortality event in 2014. GCMS, heavy metal and vitamin E levels performed on frozen liver in those cases were unremarkable. Given the hemorrhage identified in multiple infected iguanas, anticoagulant rodenticides may also be considered as a cause for mortality. However, as hemorrhage was not present in all cases and was not considered fatal in all of those that were affected, and as coagulation deficits are the only known mechanism in which these compounds contribute to death, anticoagulant intoxication was deemed unlikely. Finally, the presence of circulating Helicobacter, especially in the blood, splenic and hepatic tissue, in only sick animals supports its involvement in the disease. Similar organisms have not been seen in any pre-release blood smear evaluation since the inception of the BIRP, nor have they been seen in healthy animals held in the captive facility or opportunistically sampled free-ranging iguanas. We have added information and text to the discussion regarding potential differential diagnoses, as well as additional statements of uncertainty in the relationship between infection and mortality (beginning Line 667): 

o “A causative relationship between Helicobacter sp. GCBI1 infection, illness and/or death, as defined by Koch’s postulates, was not definitively attained in our study, as isolation of the organism has been unsuccessful thus far and therefore infection/challenge studies could not be pursued. From a clinical standpoint, the identification of Helicobacter organisms in blood smears and by PCR in sick iguanas would suggest that this organism was associated with the observed clinical signs in both iguana populations. Despite this support, the lack of disease-specific clinical signs and postmortem lesions, and the lack of consistent histopathologic identification of organisms in more common lesions, complicated confirmation of Helicobacter sp. GCBI1 as a distinct contributor to mortalities. The results of the thorough clinicopathologic approach to these mortality events described herein, however, reduce the likelihood of other principal causes and strengthen the probability that these mortalities are a result of infection by this novel Helicobacter species. Notably, neither the presence of spiral-shaped bacteria nor increased mortality levels (other than those related to known predation events) have been seen in this Grand Cayman blue iguana population in over 15 years of intensive management and monitoring. This included clinical examination and testing, including blood smears, as well as postmortem evaluation of mortalities. The relationship between the presence of Helicobacter sp. GCBI1 and disease in green iguanas was more difficult to define, as bacteremia was not adequately assessed in cases without antemortem diagnostics, and this population has not been as extensively monitored as the blue iguana population. Performing necropsies on all of the animals included in this report was a fundamental step in ruling out other causes for these events, and no other consistent disease processes were identified. Toxicologic and nutritional analyses were performed on green iguana tissues during the 2014 mortality event in the West Bay region and were unremarkable, and there was no gross or histologic evidence of toxicity or nutritional disease in the Grand Cayman blue iguanas. Finally, the response of multiple animals to antimicrobials, including clearing of organisms from circulation in the three tested iguanas, was suggestive of a bacterial cause for the illnesses and mortalities, consistent with the findings of organisms on blood smears and by PCR. While blood cultures to investigate other bacteria in circulation were not performed in sick animals (culture of blood samples was restricted to the use of Helicobacter-specific culture media), there was no indication of polymicrobial systemic infection in the blood smears examined.”

- As well as (beginning line 881):

o “Multiple avenues should be pursued as investigation of this disease progresses. Our current research shows an association between helicobacteremia and mortality in Grand Cayman blue iguanas, and an association between the presence of Helicobacter sp. GCBI1 genetic material and mortality in invasive green iguanas. The lack of specific and consistent clinicopathologic findings, however, impairs our ability to confirm the causative relationship between infection and mortality, and this is an important avenue for future research. This could include more sensitive post-mortem identification of Helicobacter in tissues and lesions such as in-situ hybridization rather than silver staining, and more comprehensive testing to better classify the negative disease state. Identification and description of a carrier state requires additional investigation. Given that we discovered the utility of choanal sampling in healthy green iguanas towards the end of our investigation, additional screening and monitoring of sympatric reptiles by multiple methods to include choanal swabs will be important to better understand the role green iguanas play in disease transmission and whether Helicobacter sp. GCBI1 may have emerged in one or both species from a crossover event. Such testing is recommended not only to identify potential healthy carriers, but to also determine Helicobacter sp. GCBI1 shedding by infected animals, treated animals before and after recovery, or during periods of physiologic or environmental stress.”

- And finally (beginning at line 922):

o “Successful culture could also allow for challenge studies to be conducted to better assess a causal relationship between infection and mortality, and better understanding of this disease from a clinicopathological perspective. Finally, additional testing to rule out other disease processes playing a role in these mortality events should be employed. While toxicology and nutritional studies were performed on green iguana tissues in 2014 with unremarkable results (heavy metal and vitamin E analysis and GC/MS), such ancillary testing was not performed on blue iguanas.”

- are the authors really sure that their Helicobacter is really intracellular? I cannot conclude this from Fig. 4, this might be an overlay phenomenon?

- While the majority of organisms are extracellular (including those overlapping erythrocytes), large numbers of densely arranged organisms are convincingly found in the cytoplasm of the mononuclear cells present. We have changed Fig 4 to a new photomicrograph which we believe to show intracellular organisms more clearly, and have added arrows to highlight the organisms (see figure caption, Lines 434-436).

- why did authors not try to find the Helicobacter OTU also in iguana food (PMID: 29457310), arthropods (PMID: 15499059) or further (iguanid) reptiles (cf. recent introduction report on Devriesea in anoles and I. delicatissima (PMID: 28970515))?

- We appreciate these suggestions and had not considered the possibility of Helicobacter in yeasts in food. Currently, the only iguanas on Grand Cayman are the greens and the blues, and we focused our non-blue iguana sampling on the greens, rather than on other reptiles. That being said, we did a small amount of testing of other reptiles (snakes, anoles, turtles) and of mosquito blood meals, but elected not to include that in this manuscript because it is a very small data set and we are hoping to broaden that investigation for a larger future study.

- concerning therapy: why not focusing on research in human EHS therapy instead of reptile formulae: PMID: 24274445 suggests macrolides and points out resistance in fluorchinolons

- This is a good point and was really an effect of the research and investigation in the US lagging behind real-time events in Grand Cayman. However, initial empiric broad spectrum treatment before the causal agent was identified resulted in iguanas surviving with this therapeutic protocol, therefore there was hesitation to change the approach once the agent was identified. It’s also notable that only antibiotics with established reptile doses were used. Although some reports do describe quinolone susceptibility, your point that most are not susceptible, and provision of this reference, is appreciated. Note that we did very briefly discuss antimicrobial usage in humans as it relates to gastric vs. enterohepatic species (Line 740-746).

Reviewer #2: This manuscript describes the characterization of a novel Helicobacter infection and its associated mortalities in endangered iguanas. The manuscript shows interesting findings but lacks however some clarity and additional results. Below an overview of my comments:

* it is difficult to extract from the text how many alive and dead animals were included in this study and to which species they belonged. This should be clarified in the text and referred to the tables.

- Thank you for pointing this out. While that information was present in the manuscript, it was not stated clearly in one location. We have adjusted lines 324-325 to clarify the numbers of live and dead animals included in this study, and now reads “In total, there were 19 Grand Cayman blue iguanas, three live and 16 dead, and three green iguanas, all dead, included in this report.” We additionally adjusted Table 1 and S1 Table to read “SURVIVED” rather than “ALIVE” (columns 5 and 7 in both tables).

*Was blood only collected from the live animals? How was Helicobacter positivity investigated in the dead animals?

- Blood was collected and banked from some live animals and from only a small amount of the “found dead” animals during the necropsy, when it was feasible (depending on length of time animal had been dead). All blood samples that were collected and imported to WCS were tested via PCR. PCR performed on tissue, both frozen and in paraffin blocks, was the primary method of investigating Helicobacter in dead animals, but silver staining was also used. This information is detailed in Table 1 and S1 Table. Additionally, more detail was added to the Methods in the pathology and molecular subsections to help clarify (lines 205-209 and lines 224 and 228, respectively).

*Also the tables are difficult to read. What is meant with non-Helicobacter sepsis? Sepsis caused in the absence of Helicobacter species or caused by non-Helicobacter pylori Helicobacter species? Does the studbook number refers to the animal number? In table 2, I would also include the animal species.

- “Non-helicobacter sepsis” was intended to indicate bacterial sepsis caused by organisms other than helicobacters. We have changed “non-helicobacter” to “bacterial.”

- Regarding the animal ID’s, the studbook number is the only method of animal identification used for the blue iguanas in this manuscript. The green iguanas are not managed and do not have studbook numbers, so their WCS necropsy number was used for identification purposes. To clarify, we added the following text to lines 147-149: “All individual blue iguanas are identified by their studbook number, and the green iguanas are identified by the necropsy number assigned to them by WCS.”

- The species common name for table 2 is included in the table title: “Table 2. Hematology results for three Helicobacter sp. GCBI1 positive Grand Cayman blue iguanas.”

*In general, Figures are of bad quality. I would recommend to upload figures with higher quality.

- The authors found that if they click on the “click here to access/download” to the upper right of the pdf file that was uploaded for review that the high quality image opens up. The figure DPI is greater than 300 and all were checked through the PACE system. We have supplied new images for Figs 1, 4 and 5, but are not sure what aspect of the images require improvement (is this referring to the digital/technical aspects of the images or the aesthetics?).

- Fig 1 has been updated to include the Cayman Islands DoE crest, as they created and own these images, and to add another location for green iguana mortalities which was inadvertently omitted originally (see fig caption, lines 108-118).

- Fig 4 has been updated as described above, with a new image showing intracellular organisms more clearly, and with arrows (see fig caption, lines 432-436).

- Fig 5 has been updated to show the spiral morphology of the bacteria more clearly and to include arrows highlighting the organisms (see fig caption, lines 574-578).

*Figure 6: phylogentic tree: why not including gastric Helicbacter species ? Does the novel Helicbacter species belongs to gastric or enterohepatic species? And are the authors convinced that this Helicobacter infection is caused by one species. Or do they think that multiple species can be involved?

- Gastric species including H. pylori, H. suis, H. bizzozeronii, H. felis and others are already included in the tree.

- We did not formally assign the novel Helicobacter as gastric or enterohepatic based solely on 16S sequence as our overall knowledge of the organism remains small and we were unable to culture the organism to understand more of its functional characteristics. We do, however, note in the manuscript that this Helicobacter falls phylogenetically within a clade of known EHS based on 16S sequence (lines 602-605). We have not detected any other species of Helicobacter in the blood or tissue samples and believe that only one organism is responsible for disease.

*Did the authors also collect environmental samples to investigate the origin of this novel Helicbacter species since it was shown by the authors that healthy iguanas are not colonized with this species in the gut?

- No environmental sampling such as water or soil was conducted for this study. However, we do briefly discuss the importance and potential for environmental DNA sampling in the discussion (lines 903-907). Culture-based environmental monitoring is a good idea for the future, but throughout this investigation, we saved our limited resources and Helicobacter culturing media for samples which were more likely to yield positive results. 

*Did the authors try to isolate the novel Helicobacter species from blood cultures? This would be of added value to further characterise this novel Helicobacter species. Does this species produces urease since it is known that some enterohepatic species have the ability to produce urease?

- Culture attempts using Helicobacter-specific media were made using blood and feces and described in lines 652-658: “Samples collected for Helicobacter culture included the 17 fecal and seven cloacal swab samples from treated and healthy Grand Cayman blue iguanas described above, blood samples from six Grand Cayman blue iguanas (Studbook #786, 2156, 537, 1046, 784, and 1278) which were positive for Helicobacter by case definition, and 45 fecal samples from the apparently healthy culled green iguanas. None of the samples produced positive cultures for this organism, including the blood samples, of which four of the six were confirmed positive for Helicobacter GCBI1 by PCR”.

- Similar to the historical difficulties with culturing H. suis (see: https://doi.org/10.1099/ijs.0.65133-0), we have so far been unsuccessful in isolating this organism in culture.

*page 22 (bottom): one animal, that survived the infection, showed a year after the same clinical signs but was now negative for Helicobacter. What was the cause then? Does this questions the pathogenic role of this Helicobacter species in iguanas?

- This situation was also questioned by the first reviewer, to which we responded: “Studbook #2156 initially presented on 12Jun2016 and was positive for helicobacter via blood smear and blood PCR at that time. She recovered and then re-presented with similar signs, although diagnostics did not reveal Helicobacter infection. Unfortunately, the only Helicobacter testing performed at that time was a blood smear, which was read as negative. As the clinical signs are not specific for Helicobacter infection, it is possible that this was an unrelated illness and I apologize for any suggestion that this was a relapse. As such, and since this re-presentation would not have qualified for inclusion in this manuscript based on our parameters, we have removed the following sentence from the manuscript (lines 511-514): “One of the surviving animals (Studbook No. 2156) presented one year after the initial episode, with the same clinical signs as a year before. Both hospitalization and recovery times were shorter, with two days of hospitalization in 2017 vs eight days in 2016. However, infection with Helicobacter sp. GCBI1 was not confirmed in 2017.”” 

- In short, this could have been an unrelated illness, but without additional diagnostics, we cannot comment in detail on this re-presentation.

*How would you explain the fact that Helicobacter GCBI1 was only found in blood smears but not in affected organs upon pathological examination? Would this again questions the pathogenic role in this animal species?

- A great question. Bacterial sepsis can lack identifiable organisms on histology as much of the septic process is the result of the systemic inflammatory response rather than the organism itself (see doi: 10.1007/1-84628-026-5_3). Additionally, silver stains are notoriously difficult to read/interpret because of overstaining (too much artefact) or understaining which make it difficult to highlight bacteria in tissue samples. If bacteria are present in tissue in low numbers, detection becomes a real issue, which is where PCR can be helpful. A comment to this effect has been added at Lines 886-889: “This could include more sensitive post-mortem identification of Helicobacter in tissues and lesions such as in-situ hybridization rather than silver staining, and more thorough testing to better classify the negative disease state.”

ADDITIONAL COMMENTS IN TRACKED CHANGES IN THE PDF “PONE-D-20-22644_reviewer” ARE ADDRESSED BELOW (Reviewer comment in BOLD with PDF line number):

Title: Please check with respect to actual taxonomy changes within the genus Iguana 

Iguana iguana is the currently listed taxonomic designation for the green iguana per ITIS, website accessed on 27Oct20, and on IUCN, website accessed on 27Oct20.

Line 47: Is there any evidence for a transmission from that place? which tissues? 

We don’t have any direct evidence of transmission from the green iguana mortality event to blue iguanas, and we state in line 52 of the track changes document that “The source of infection and mode of transmission are yet to be confirmed.” Tissues which tested positive in green iguanas from that event are listed in S1 Table. Additionally, there are no blue iguanas in the regions of Grand Cayman where the green iguana mortality event occurred.

Line 81: Reviewer corrected a typo in Campylobacterales

This has been corrected in the track changes version line 86

Line 84: helicobacteriosis

This has been corrected to helicobacteriosis in the track change version line 89

Line 87: please specify 

“and Dinosauria” has been removed in the track change version line 92; while we were referring to birds, they are actually not mentioned in this reference (8), and their inclusion is extraneous to this manuscript. Thank you for identifying this.

Line 99: based on my comments below I would rather prefer to talk about a GCBI1-specific PCR

We have changed “pathogen-specific” to “helicobacter-specific” PCR assay (line 105 in track changes).

Line 103: Reviewer highlighted helicobacter

Helicobacter (capitalized and italicized) has been corrected throughout the manuscript in the track changes document.

Line 127: “pathogenic” is highlighted, but without comment

Authors are assuming the reviewer suggested the removal of this word which we have changed to “target” (track changes line 140). 

Line 168-169: text highlighted but not comments

Authors are unsure of the significance of highlighting.

Line 198: Please add more details here in a separate chapter Bacteriology.

A separate “Bacteriology” paragraph has been added to the methods (lines 290-294 in the track changes document) and lines 651-658 in the results sections of the track changes document.

Line 302, Table 1: Inappropriate table format cannot fully be evaluated. Furthermore, footnotes and abbreviations (M, F) have to be properly introduced.

I’m very sorry that this table was not legible in the pdf version, but it apparently should have been available in the draft view of the word document per the PLOS author instructions (https://journals.plos.org/plosone/s/tables): “Tables do not have strict width and height requirements. Do not split your table or otherwise try to make the table appear within the manuscript margins if it does not fit on one page. In Word, tables that run off of the manuscript page can be seen using Draft View. In the PDF version of the published article, very wide tables may be printed sideways, and long tables may span more than one page.” Please advise if this table needs to be adjusted for publication.

M, F and U have now been defined in the footnotes of Table 1 and S1 Table.

Table 1: Not sure, but normally symptoms is restricted to humans, so maybe better: Date of onset of disease signs?

Symptom has been changed to “Date of Onset of Clinical Signs” in column 4 of Table 1 and S1 Table.

Table 1: Please specify the ID method (qPCR?) here again. I would prefer positive and negative instead.

“Helicobacter positive” has been changed to “Helicobacter positive per case definition” in the sixth column of Table 1 and S1 Table to reflect that this was an overall identification of positive state by any means within our defined criteria (methods); because of this, YES and NO remained in the table.

Table 1: The word “Open” is crossed out

Open has been changed to inconclusive in Table 1 and S1 Table as cause of death to improve clarity.

Line 318: why in a deviant position compared to M&M?

The Climatology section in methods has been reordered to line 151 in the track changes version to match the order in results.

Line 327: why not giving wheather reports for the whole year? One can wonder whether there might be a second phase with higher precipitation?! 

We focused on the time frame in which Helicobacter-related mortalities occurred, as we were hoping to identify patterns related to illness. A comment to this effect has been added to the caption for Fig2 (lines 379-384 in track changes): “Precipitation measurements in mm (green bars) for the April-October periods for a) 2015 and b) 2016. Dates of Grand Cayman blue iguana onset of signs attributed to Helicobacter infection or dates Helicobacter positive specimens were found deceased are designated by red lines. For the ease of visualization, presented are the precipitation measurements bracketing the time period when the Helicobacter-related mortalities occurred.”

Line 363: Why did this not lead into a more profound neurological examination, also with the signs listed below and also in the following year?

We attributed these signs to general weakness rather than true neurologic deficits, but agree that the CNS should have been better evaluated histologically to further investigate this. In almost all cases, the central nervous system was not collected postmortem. In those cases in which brain was available for examination, it was unremarkable. We have added a sentence regarding tissue collection to the methods (lines 205-209 in track changes): “Collected tissue sets were inconsistent, but nearly all cases included at least liver, lung, kidney, stomach, intestine and skeletal muscle in formalin (one case, Studbook No. 3003 was markedly autolyzed and only skeletal muscle and skin were collected); central nervous system tissues were infrequently included. Frozen tissue collection varied case-to-case.”

As mentioned in response above, physical examinations were performed on all blue iguanas presenting clinically ill, which included assessment of the neurologic status, but we failed to specifically mention that in the manuscript. We have added the following comment (lines 415-417 in track changes): “Specific neurologic deficits, beyond depression and stupor, were not noted on physical examination in any of the animals (e.g. no proprioception deficits, ataxia, nystagmus, head tilt, etc.).”

Table 2: same problem with this and the following table...

Please see my prior note pertaining to Table 1 and formatting.

Lines 401 and 402: Symptom is highlighted in two places.

The text in lines 450-452 in track changes has been adjusted to read: “In two cases (Studbook No. 895 died 5 days after the onset of clinical signs and Studbook No. 1046 died 17 days after the onset of clinical signs)…” Symptom has been changed.

Line 422: Can this and some of the "improved" values listed above be attributed to your supporting care management, e.g. the infusion mentioned below?

Yes, the PCV and protein levels can be negatively affected by fluid therapy due to dilution, appearing similar to changes caused by hemorrhage. However, the decreases seen in at least two of the three animals with progressively decreasing PCV levels showed a marked reduction that surpassed what would be expected from fluid therapy alone. A comment to this effect has been added (lines 473-475 in track changes): “Two of the three follow up PCV measurements decreased by greater than 50% over two days, which we considered too great to attribute solely to the dilution effect of fluid therapy.”

Line 470: it is 18 days above

Thank you for mentioning this, it has been corrected in the footnotes of Table 1 and Supporting information Table S1 (17 days is correct): “Blue iguana 1046 had spirilliform bacteria on blood smear at her initial presentation; she improved initially, but died after a short second illness 17 days later.”

Line 549-550: Duplication

We have corrected lines 608-611 in track changes to remove the duplication and it now reads: “While Helicobacter sp. GCBI1 was found in the blood of clinically ill animals, Helicobacter sp. MIT 16-1353 was only identified in feces and was not associated with septicemia.”

Figure 6: I doubt that quality of Fig. 6 is sufficient

We will work to ensure the quality of this image is to the standard of PloS One guidelines; note that it has a resolution of 600dpi and was cleared by the PACE system prior to submission. We found that if the link on the upper right of the image is clicked, a full resolution version is downloaded and is much higher quality than the version presented in the pdf figure.

Lines 743-753: The whole paragraph is rather speculative and can be ommmitted here.

This paragraph has been omitted in the track change version at line 847.

---

## [Editor Report · Decision Letter 1]

1 Feb 2021

Systemic *Helicobacter* infection and associated mortalities in endangered Grand Cayman blue iguanas (*Cyclura lewisi*) and introduced green iguanas (*Iguana iguana*)

PONE-D-20-22644R1

Dear Dr. Conley,

We’re pleased to inform you that your manuscript has been judged scientifically suitable for publication and will be formally accepted for publication once it meets all outstanding technical requirements.

Kind regards,

Patrick Butaye, DVM, PhD

Academic Editor

PLOS ONE
---

## [Editor Report · Acceptance letter]

8 Feb 2021

PONE-D-20-22644R1 

Systemic *Helicobacter* infection and associated mortalities in endangered Grand Cayman blue iguanas (*Cyclura lewisi*) and introduced green iguanas (*Iguana iguana*) 

Dear Dr. Conley:

I'm pleased to inform you that your manuscript has been deemed suitable for publication in PLOS ONE. Congratulations! Your manuscript is now with our production department. 

Kind regards, 

on behalf of

Professor Patrick Butaye 

Academic Editor

PLOS ONE